# Certifying Graph Neural Networks Against Label and Structure Poisoning

Lukas Gosch [* 1 2 3]   Xichuan Chen [* 1]   Yan Scholten [* 1 2 3]   Stephan Günnemann [1 2 3]

## Abstract

Robust machine learning for graph-structured data has made significant progress against test-time attacks, yet certified robustness to poisoning – where adversaries manipulate the training data – remains largely underexplored. For image data, state-of-the-art poisoning certificates rely on partitioning-and-aggregation schemes. However, we show that these methods fail when applied in the graph domain due to the inherent label and structure sparsity found in common graph datasets, making effective graph-partitioning difficult. To address this challenge, we propose a novel semi-supervised learning framework called deep Self-Training Graph Partition Aggregation (ST-GPA), which enriches each graph partition with informative pseudo-labels and synthetic edges, enabling effective certification against node-label and graph-structure poisoning under sparse conditions. Our method is architecture-agnostic, scales to large numbers of partitions, and consistently and significantly improves robustness guarantees against both label and structure poisoning across multiple benchmarks, while maintaining strong clean accuracy. Overall, our results establish a promising direction for certifiably robust learning on graph-structured data against poisoning under sparse conditions.

## 1. Introduction

Graph Neural Networks (GNNs) are highly susceptible to adversarial perturbations in their input graph applied at test or training time (Zügner et al., 2018; Dai et al., 2018). Subsequently developed empirical defenses are at the continual risk of being rendered ineffective by more sophisticated ways to choose adversarial perturbations (Koh et al., 2022; Mujkanovic et al., 2022). This motivates the development of robustness certificates, which provide provable guarantees about the stability of predictions under worst-case data perturbations, allowing to rigorously assess and mitigate adversarial vulnerabilities. While significant advances in providing such provable guarantees for GNNs against test-time attacks have been made (Günnemann, 2022; Hojny et al., 2024), certifying robustness of GNNs against data poisoning, where an adversary can manipulate the graph structure (Zügner & Günnemann, 2019) or node-labels (Lingam et al., 2024) at training time, remains largely underexplored.

The most effective approaches to derive poisoning robustness guarantees in the image domain rely on partitioning the training data, learning separate (base) classifiers, and aggregating predictions (e.g., via majority vote) (Levine & Feizi, 2021). However, we demonstrate that these methods fail when directly applied to common graph learning tasks such as node classification. In particular, we find that the core challenge is *sparsity*: both labels and graph structure are often too sparse to provide effective training signals under data partitioning, leading to poor performance for label certificates and vacuous robustness guarantees for structure poisoning. This raises a critical question:

> *How can we effectively analyze and guarantee the trustworthiness of GNNs in the presence of structure and label poisoning, while maintaining their utility?*

In this work, we address this challenge by introducing a novel semi-supervised learning framework called deep Self-Training Graph Partition Aggregation (ST-GPA) that successfully overcomes the sparsity problem and enables effective poisoning certification of GNNs against structure and label poisoning (see Fig. 1). In particular, our method enriches the subgraphs created through partitioning the original graph, with synthetic data generated using carefully designed self-training approaches. *Self-training* is a concept from semi-supervised learning (Chapelle et al., 2006) that refers to training a model on its own predictions to expand the leverageable training set. Concretely, for each sparsely labeled graph partition, our method efficiently generates synthetic (pseudo) labels, expanding the available node-label set. In a similar spirit, for each sparsely connected subgraph, our method generates synthetic edges through

---

[*]Equal contribution . [1]School of Computation, Information and Technology, Technical University of Munich [2]Munich Data Science Institute (MDSI), Technical University of Munich [3]Munich Center for Machine Learning (MCML). Correspondence to: Lukas Gosch <l.gosch@tum.de>.

*Proceedings of the 43rd International Conference on Machine Learning*, Seoul, South Korea. PMLR 306, 2026. Copyright 2026 by the author(s).

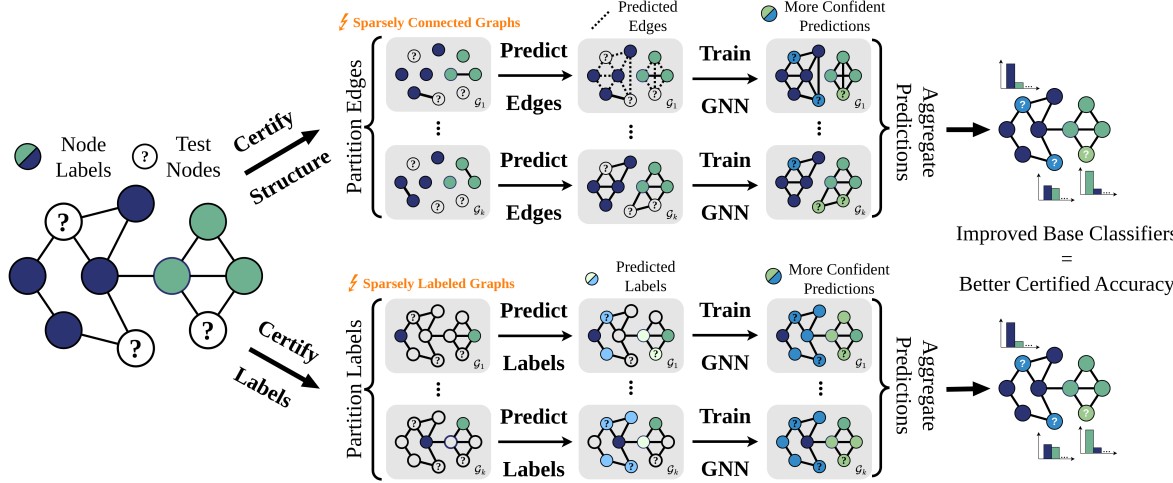

*Figure 1.* Deep Self-Training Graph Partition Aggregation (ST-GPA). To certify against *structure poisoning*, the graph's edges are partitioned. The resulting set of sparse graphs are enriched with informative pseudo-edges and then, an ensemble of GNNs is trained on the partitions. To certify against *label poisoning*, the graph's node labels are partitions, and the resulting sparsely labeled graphs enhanced with pseudo-labels, before training the ensemble. To *jointly certify* against both poisoning, ST-GPA partitions both the edges and labels and adds pseudo-edges and pseudo-labels.

solving a link prediction problem, endowing each subgraph with a more informative structure. Then, by training GNNs as base classifiers on these enriched subgraphs and aggregating their outputs using a majority vote, we obtain strong certificates for label poisoning, structure poisoning, and joint label-structure poisoning. Our contributions are:

**(i)** In Sec. 3 we generalize the partition-and-aggregate paradigm prevalent in the image domain to derive poisoning robustness certificates to non-i.i.d structured data. Then, in Sec. 4, we **identify and characterize a failure mode** of certifying robustness against poisoning using this paradigm **on graph-structured data** rooted in the label and structure sparsity found in common graph datasets.

**(ii)** In Sec. 5 we propose deep Self-Training Graph Partition Aggregation (ST-GPA), a novel semi-supervised learning framework that augments partitions with carefully obtained pseudo-labels and/or synthetic edges, **overcoming the sparsity problem** when partitioning graph-structured data.

**(iii)** ST-GPA is the first effective certificate against structure and label poisoning for node classification with graph neural networks. In particular, we set a **new state-of-the-art certified poisoning robustness for GNNs** across multiple benchmarks, while being efficient and obtaining competitive clean accuracies (see Sec. 6).

Overall, our work highlights the importance of self-training in certifying poisoning robustness of GNNs and lays the foundation for future work against training-time attacks in the graph domain. We believe that our insights on improving partition-based certification using self-training may be of independent interest beyond graph learning.

## 2. Preliminaries

We consider an attributed, undirected graph $G \subseteq \mathbb{G}$ described by a feature matrix $\boldsymbol{X} \in \mathbb{R}^{n \times m}$ of $m$-dimensional features for $n$ nodes and a set of edges $\mathcal{E} \subseteq \{\{i,j\}, i,j \in [n]\}$ with $[n] = \{1, \ldots, n\}$, where two nodes $i$ and $j$ are connected if and only if $\{i,j\} \in \mathcal{E}$, and each node belongs to one out of $C$ classes with $\mathbb{Y} = [C]$. We study semi-supervised transductive node classification. That is, the full graph $G = (\mathcal{E}, \boldsymbol{X}, \boldsymbol{y})$ is available during training but its nodes are only partially labeled. This is modeled with a label vector $\boldsymbol{y} \in (\mathbb{Y} \cup \{-1\})^n$, where entry $y_i \in \mathbb{Y} = [C]$ indicates the label of node $i$, and $y_i = -1$ means the node $i$ is unlabeled. We model GNNs as functions $f : \mathbb{G} \to \mathbb{Y}^n$, and collect the labels in the graph in the set $\mathcal{Y} = \cup_{v \in V_{\text{labeled}}} \{(v, \boldsymbol{y}_v)\}$ and the node attributes in the set $\mathcal{X} = \cup_{v \in V} \{(v, \boldsymbol{X}_v)\}$. We write graph datasets as $D = (\mathcal{E}, \mathcal{X}, \mathcal{Y})$.

**Perturbation model.** In this work we study two types of poisoning attacks on graph-structured data: *label flipping* and *structure perturbations*. To quantify the strength of an attack, we define *attack budgets* $r_l$, $r_s$, which bound the number of allowed modifications to labels and structure, respectively. We model the set of possible poisoned graphs $\tilde{G}$ as a ball centered around the clean graph $G$:

$$\mathcal{B}_{r_l, r_s}(G) = \big\{ \tilde{G} = (\tilde{\mathcal{E}}, \boldsymbol{X}, \tilde{\boldsymbol{y}}) \mid \delta(\tilde{\boldsymbol{y}}, \boldsymbol{y}) \le r_l, \\ \Delta(\tilde{\mathcal{E}}, \mathcal{E}) \le r_s \big\} \quad (1)$$

where $\Delta(\mathcal{X}, \mathcal{Y}) = |(\mathcal{X} \setminus \mathcal{Y}) \cup (\mathcal{Y} \setminus \mathcal{X})|$ is the symmetric difference of two sets, and $\delta(\tilde{\boldsymbol{y}}, \boldsymbol{y}) = \sum_{i=1}^n \mathbf{1}_{\tilde{y}_i \ne y_i}$ is the number of different entries (Hamming distance) of two vectors. Eq. (1) models three different perturbation models: (i) label flipping ($r_l > 0, r_s = 0$); (ii) structure poisoning

$(r_l = 0, r_s > 0)$; and (iii) both together $(r_l > 0, r_s > 0)$.

**Deep partition aggregation.** State-of-the-art certified robustness guarantees for supervised image classification under poisoning attacks have been achieved via the partition-and-aggregate paradigm. The most prominent implementation of which is Deep Partition Aggregation (DPA) (Levine & Feizi, 2021). The core idea is to partition an i.i.d. training set $D$ into $k$ disjoint subsets and to train independent classifiers $f_D^{(i)} : \mathcal{X} \to \mathcal{Y}$ deterministically on each partition $i$. At inference time, for a given input $x \in \mathcal{X}$, the final prediction is made via *majority vote* across all base classifiers:

$$g_D(x) = \arg\max_{c \in \mathcal{Y}} n_c(D, x), \qquad (2)$$

where $n_c(D, x) = \sum_{i=1}^{k} \mathbf{1}[f_D^{(i)}(x) = c]$ counts the number of classifiers predicting class $c$ for input $x$. This setup enables a formal robustness guarantee for the aggregated prediction under poisoning:

**Theorem 2.1** (Levine & Feizi (2021)). *Given a clean dataset $D$, the majority-vote prediction remains unchanged, i.e. $g_D(x) = g_{\tilde{D}}(x)$, for any perturbed dataset $\tilde{D} \in \mathcal{B}_r(D)$ bounded by the attack budget $r$, as long as*

$$r \leq \left\lfloor \frac{n_c(D, x) - \max_{c' \neq c} (n_{c'}(D, x) + \mathbf{1}_{c' < c})}{2} \right\rfloor, \quad (3)$$

*where $c = g_D(x)$ is the predicted class on the clean dataset.*

## 3. Generalized Deep Partition Aggregation for Non-i.i.d. Data

Even though DPA has been formulated for image data, the general paradigm to derive poisoning guarantees can be readily generalized to non-i.i.d. structured data. The main idea is based on recognizing that deriving a guarantee like Thm. 2.1 does not depend on the i.i.d. nature of the dataset, but rather on a partitioning scheme $h$ where the potentially poisoned objects are partitioned into subsets *independent* of one another. In particular, it can be formulated w.r.t. a general set of objects $\mathcal{O}$ that may be poisoned, where $o_1 \in \mathcal{O}$ may not be independent of another $o_2 \in \mathcal{O}$ (e.g., in a graph, one node may not be independently sampled from another node), as long as the partition $o_1$ is grouped into, is not affected by the value of any other element in $\mathcal{O}$, and thus, poisoning any object can only affect one partition.

To formulate such a split-and-majority voting certificate for general (potentially structured) data, assume a dataset $D = (\mathcal{T}, \mathcal{O})$ consisting of a set of objects $\mathcal{T}$ that are known to be clean and a set of objects $\mathcal{O}$ that may be poisoned. For example, for i.i.d. image data $\mathcal{O} = \{x_i\}_{i=1}^{n}$ and $\mathcal{T} = \varnothing$. Then, $k$ partitions $\mathcal{P}_i$ with $i \in [k]$ are created as follows:

$$\mathcal{P}_i := (\mathcal{T}, \{o \in \mathcal{O} \mid h(\mathcal{T}, o) = i \pmod{k}\}) \quad (4)$$

where $h$ is a deterministic (hash) function that takes $\mathcal{T}$ and one $o \in \mathcal{O}$ as input and outputs a number in $\mathbb{N}$ representing the partition index, into which the given object $o$ should be grouped. Then, one independently trains base classifiers $f_D^{(i)} : \mathbb{X} \to \mathbb{Y}$ on each partition $\mathcal{P}_i$, where $\mathbb{X}$ represents a general data domain. Without loss of generality, we assume $f_D^{(i)}$ to output a scalar class prediction, i.e., $\mathbb{Y} \subseteq \mathbb{N}$. For example, for node classification, the data domain can be defined as $\mathbb{X} = (\mathbb{G}, \mathbb{N}_0)$ where $\mathbb{G}$ is the set of possible graphs and the second element in the tuple refers to the index of a node, for which a class prediction is sought.[1] Now, given an input $x \in \mathbb{X}$, the final prediction is made as in DPA for images via a majority vote across all base classifiers (Eq. (2)), which we denote $g_D(x)$.

Lastly, for general data domains, the set difference $\Delta(D, \tilde{D})$ may not always be an appropriate distance measure. Thus, assume a general distance function $d(D, \tilde{D})$ between the clean dataset $D$ and the perturbed dataset $\tilde{D}$. Further, different choices of hash function $h$ may affect how many partitions $d_h \in \mathbb{N}$ are at most affected when poisoning an object $o \in \mathcal{O}$. If a perturbation of size $d(D, \tilde{D})$ leads to at most $p$ changed partitions given $h$. Then, the scalar $d_h$ links $d(D, \tilde{D})$ to $p$ as follows: $d_h \cdot d(D, \tilde{D}) \geq p$. Exemplary, $d_h > 1$ if one object is partitioned (duplicated) into multiple partitions as done by Wang et al. (2022). Now, we can state the following general theorem that follows the proof strategy outlined by Levine & Feizi (2021) and we refer to App. C.1 for the proof:

**Theorem 3.1** (Generalized DPA). *Given a clean, possible non-i.i.d and structured dataset $D$, and a poisoned dataset $\tilde{D}$, the majority-vote classifier prediction remains unchanged, i.e., $g_D(x) = g_{\tilde{D}}(x)$, as long as $d_h d(D, \tilde{D}) \leq \lfloor (n_c(D, x) - \max_{c' \neq c} (n_{c'}(D, x) + \mathbb{1}_{c' < c}))/2 \rfloor = r_m(D, x)$, where $c = g_D(x)$ is the predicted class on the clean dataset.*

We refer to the right-hand side of the condition in Thm. 3.1 as the *robust margin $r_m(D, x)$* of a sample $x$ given a dataset $D$. To conclude, we need to find a function $h$, a scalar $d_h$, and a distance measure $d(D, \tilde{D})$ to get a robustness guarantee for an ensemble classifier with Thm. 3.1.

**Generalized DPA for graphs.** We define a general hash function in Eq. (5) that allows to partition a graph based on its node features $\mathcal{X}$. This hash function can be used for all three perturbation models captured by Eq. (1):

$$h(\mathcal{X}, o) = \begin{cases} h(X_i || X_j) + h(X_j || X_i) & \text{if } o \in \mathcal{E} \\ h(X_v) & \text{if } o \in \mathcal{Y} \end{cases} \quad (5)$$

The hash function $h(\mathcal{X}, o)$ determines the partition index for an edge, by taking the features of the incident nodes

---

[1] If $f_D^{(i)}$ outputs a vector for several indexable objects (e.g., nodes in a graph), this can be equivalently represented as a scalar prediction for each indexable object, where the index of the object for which the prediction is sought for, is part of $\mathbb{X}$.

and concatenating them. We add both orderings to make the partitioning process invariant to the actual node order. When partitioning labels, $h(\mathcal{X}, o)$ takes the corresponding node's features as input. The $h$ in Eq. (5) can be any hash function, in our experiments we choose the MD5 hash. As each edge or label is in exactly one partition, $d_h = 1$.

Finally, we choose three distance metrics for the three different threat models. For **label poisoning**, we take the Hamming distance $\delta(\boldsymbol{y}, \tilde{\boldsymbol{y}})$ between label vectors. For **graph structure poisoning**, we use the symmetric set difference $\Delta(\mathcal{E}, \tilde{\mathcal{E}})$ between edge sets. Inserting or deleting an edge in the graph leads to $\Delta(\mathcal{E}, \tilde{\mathcal{E}}) = 1$ and the modification of an edge to $\Delta(\mathcal{E}, \tilde{\mathcal{E}}) = 2$. Similarly, the insertion or deletion of an edge leads to exactly one affected partition based on $h$, but a modification of an edge affects two. For a **combination of label and structure poisoning**, we add up the two distances $d((\mathcal{E}, \boldsymbol{y}), (\tilde{\mathcal{E}}, \tilde{\boldsymbol{y}})) = \delta(\boldsymbol{y}, \tilde{\boldsymbol{y}}) + \Delta(\mathcal{E}, \tilde{\mathcal{E}})$. This is indeed a distance metric (see App. C.2). Given the graph dataset $D$ from Sec. 2, plugging in these distance measures along with our $h$ from Eq. (5) and $d_h = 1$ into Thm. 3.1, we get the following conditions for our certificates to hold:

$$\tilde{G} \in \mathcal{B}_{r_l>0, r_s=0}(G) : \delta(\boldsymbol{y}, \tilde{\boldsymbol{y}}) \le r_m(D, x) \tag{6}$$

$$\tilde{G} \in \mathcal{B}_{r_l=0, r_s>0}(G) : \Delta(\mathcal{E}, \tilde{\mathcal{E}}) \le r_m(D, x) \tag{7}$$

$$\tilde{G} \in \mathcal{B}_{r_l>0, r_s>0}(G) : d((\mathcal{E}, \boldsymbol{y}), (\tilde{\mathcal{E}}, \tilde{\boldsymbol{y}})) \le r_m(D, x) \tag{8}$$

where $r_m(D, x)$ is the robust margin introduced in Thm. 3.1. In the context of transductive node classification, $x$ refers to a node that we seek to classify in the graph $D$. In general, $x$ could refer to a node in an arbitrary graph different to $D$.

## 4. Limitations of the Simple Partitioning in the Graph Domain

We first expose that the partitioning scheme used by DPA does not work well on graph datasets due to their inherent label and structure sparsity. In the context of image classification, obtaining provable poisoning robustness through partitioning yields significant results (Levine & Feizi, 2021). For example, using 1,000 partitions on CIFAR-10 achieves a certified accuracy of 50% against 392 label flips. The key factor in deriving strong robustness guarantees through partitioning is the number of partitions $k$. Increasing $k$ increases the possible perturbation budget one can certify. Notably, the maximum number of tolerated perturbations before the certified ratio drops to 0% is $\lfloor \frac{k}{2} \rfloor$. Thus, one can argue that the number of partitions $k$ should be as large as possible to produce good certified radii. The primary constraint on increasing $k$ is the size of the training dataset. With 50,000 training images in CIFAR-10, it is feasible to use $k = 1,000$ partitions while maintaining acceptable performance of the base classifiers $f_D^{(i)}$. Critically, this approach does not straightforwardly translate to graph learning as we outline in the following.

Commonly used semi-supervised graph datasets come with significantly lower label rates compared to image datasets (see Table 1 in App. A). Regarding label-flipping, applying the partition-

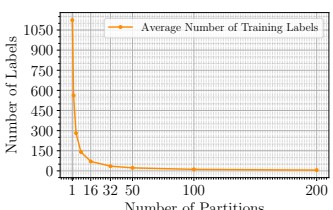

*Figure 2.* Label sparsity (Cora-ML)

ing scheme naively to node labels results in partitions with few labeled nodes – often fewer than the number of classes. Fig. 2 shows that when using 40% of nodes for training on Cora-ML, having $k = 281$ partitions yields an average of only 4 labeled nodes per partition, even though there are 7 classes. This restricts the certification bound to 140 label flips, and base classifiers suffer from significantly reduced performance, as each model only has access to on average 4 labels during training. Even worse, Fig. 4(a) shows that $k = 80$ partitions already leads to a clean accuracy of only close to 40% for an ensemble of GCNs (Kipf & Welling, 2017) on Cora-ML, whereas a GCN trained without partitioning achieves ∼78.77% accuracy.

For structure partitioning, the graph structure rapidly deteriorates as edges are divided among partitions, rendering the approach ineffective for GNNs. This is illustrated in

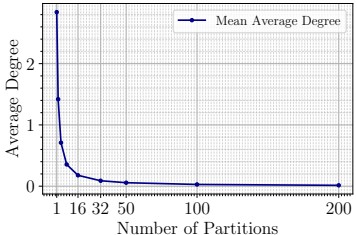

*Figure 3.* Structure sparsity (Cora-ML)

Fig. 3, where we plot the mean of average node degrees across partitions. When the number of partitions exceeds $k > 50$, connectivity in the different subgraphs becomes virtually nonexistent. Since GNNs rely on message passing to aggregate information from neighbors, this severely impairs their ability to learn meaningful representations. Under such conditions, the classifiers' performance deteriorates to that of a multilayer perceptron (MLP), which, however, is already robust to structure perturbations since it does not utilize any edge information. Overall, partitioning-based robustness certificates cannot be naively applied in the graph domain due to the inherent label and structure sparsity, motivating the need for more sophisticated methods.

## 5. Deep Self-Training Graph Partition Aggregation

To address these sparsity challenges, we propose various semi-supervised learning methods to enhance the performance of the weak classifiers by selecting either pseudo-labels or edges, or both, on each partition's limited training data. This enables us to obtain strong certificates on graphs.

**Algorithm 1** ST-GPA against label flipping

**Input:** Graph data $D = (\mathcal{E}, \mathcal{X}, \mathcal{Y})$, selectiveness $t$, co-training method $\text{CT}(\mathcal{E}, \mathcal{X}, \boldsymbol{y}, t)$, self-training method $\text{ST}(\mathcal{E}, \mathcal{X}, \boldsymbol{y}, t)$, training order $o_{trn} = (op_1, \ldots, op_R) \in \{\text{CT}, \text{ST}\}^R$, hash function $h$, number of partitions $k$

**Output:** An ensemble classifier prediction and the associated certificate for a test node $v$.

1: Split labels into partitions
 $\boldsymbol{y}_i = \{y | y \in \mathcal{Y}, h(\mathcal{X}, y) \equiv i \pmod{k}\}$
2: **for** each partition $i$ **do**
3:     **for** each operation $op$ in given order $o_{trn}$ **do**
4:         $\hat{\boldsymbol{y}} = op(\varepsilon_i, \mathcal{X}, \boldsymbol{y}_i, t)$
5:         $\boldsymbol{y}_i = \boldsymbol{y}_i \cup \hat{\boldsymbol{y}}$
6:     **end for**
7:     Train $f_i$ on $\hat{D} = \{\mathcal{E}, \mathcal{X}, \boldsymbol{y}_i\}$
8: **end for**
9: Count base classifier predictions $n_c(v)$ for all $c \in [C]$
10: Obtain majority class $c^* = \arg\max_{c \in [C]} n_c(v)$
11: Calculate robust margin (see Thm. 3.1)
 $r_m = \lfloor (n_{c^*}(v) - \max_{c' \neq c^*} (n_{c'}(v) + \mathbb{1}_{c' < c^*}))/2 \rfloor$
12: **return** $c^*, r_m$

**Algorithm 2** ST-GPA against structure perturbations

**Input:** Graph dataset $D = (\mathcal{E}, \mathcal{X}, \mathcal{Y})$, selectiveness $\varepsilon$, link prediction method $\text{LP}(e_i, \mathcal{X}, \mathcal{Y}, \varepsilon)$, hash function $h$, number of partitions $k$

**Output:** An ensemble classifier prediction and the associated certificate for a test node $v$.

1: Split edges into partitions
 $e_i = \{e | e \in \mathcal{E}, h(\mathcal{X}, e) \equiv i \pmod{k}\}$
2: **for** each partition $i$ **do**
3:     $\hat{e}_i = \text{LP}(e_i, \mathcal{X}, \mathcal{Y}, \varepsilon)$
4:     $e_i = e_i \cup \hat{e}_i$
5:     Train $f_i$ on $\hat{D} = \{e_i, \mathcal{X}, \mathcal{Y}\}$
6: **end for**
7: Count base classifier predictions $n_c(v)$ for all $c \in [C]$
8: Obtain majority class $c^* = \arg\max_{c \in [C]} n_c(v)$
9: Calculate robust margin (see Thm. 3.1)
 $r_m = \lfloor (n_{c^*}(v) - \max_{c' \neq c^*} (n_{c'}(v) + \mathbb{1}_{c' < c^*}))/2 \rfloor$
10: **return** $c^*, r_m$

## 5.1. Semi-Supervised Learning for Label Generation

To maximize utilization of the limited available labels within each partition, we employ two complementary pseudo-label generation methods: (i) Co-Training (CT), and (ii) Self-Training (ST). *First*, co-training leverages graph structure to propagate existing labels to neighboring nodes. While Li et al. (2018) proposes to generate pseudo-labels for the training of a GNN using ParWalks (Wu et al., 2012), its high computational complexity renders this approach impractical for larger graphs. Instead, we propose the use of label propagation (LP) (Zhu & Ghahramani, 2003) to significantly speed up the co-training process. The key advantage of LP is that it does not require computing the inverse of a Laplacian matrix inherent to ParWalks, and thus can be efficiently applied to larger graphs. *Second*, for self-training, we propose training a GNN on the existing labeled data and selecting the most confident predictions (as determined by their softmax scores) to serve as pseudo-labels for subsequent training iterations. Notably, both methods (co- and self-training) can be applied consecutively as we demonstrate in Sec. 6, where we observe that applying first co-training and then self-training works best (see Fig. 5(a)).

In LP we propagate a score matrix $\boldsymbol{S} \in \mathbb{R}^{n \times c}$ in which $S_{i,j}$ represents the likelihood of node $i$ belonging to class $j$. $\boldsymbol{S}$ is initialized as one-hot encoded label matrix $\boldsymbol{S}^{(0)} = \boldsymbol{Y}$ (where an unlabeled node has a row of zeros in $\boldsymbol{Y}$) and propagated with the normalized adjacency matrix $\tilde{\boldsymbol{A}} = \boldsymbol{D}^{-\frac{1}{2}} \boldsymbol{A} \boldsymbol{D}^{-\frac{1}{2}}$, as shown in Eq. (9):

$$\boldsymbol{S}^{(i+1)} = \alpha \tilde{\boldsymbol{A}} \boldsymbol{S}^{(i)} + (1 - \alpha) \boldsymbol{Y} \qquad (9)$$

where $1 - \alpha$ is the teleport probability. This propagation is iterated until convergence or a cutoff iteration. Both co-training and self-training result in a score matrix $\boldsymbol{S} \in \mathbb{R}^{n \times c}$, which is used for selecting pseudo-labels. We introduce a hyperparameter $t$ representing the number of pseudo-labels with the highest scores added *per class* via either method to control the selectiveness. A lower $t$ adds less but often higher-quality pseudo-labels, while higher $t$ adds more but less-confident ones. We present the full certification pipeline against label flipping in Algo. 1. Note that only Lines 9-11 have to be repeated to get a certificate for other nodes.

## 5.2. Semi-Supervised Learning for Edge Generation

Motivated by the concept of self-training in graph machine learning, we extend semi-supervised learning to edge prediction by generating pseudo-edges instead of pseudo-labels. Guided by the homophily assumption, we generate pseudo-edges that connect node pairs likely belonging to the same class based on a model trained on the current graph. Here, modern link prediction methods (Zhang & Chen, 2018; Kazi et al., 2023) struggle as the edges in every partition are extremely sparse (see Sec. 4). Instead, we try to capture the overall graph structure and dilute the effect of possible inter-class edges by adding in a denser graph than the original one. Within each partition, we first train a GNN to obtain initial node predictions and confidence scores, typically represented by the softmax outputs of the final GNN layer. We then iteratively add edges between node pairs with the highest sum of confidence of being in the same class. The number of edges added is controlled by a hyperparameter $\varepsilon$, which specifies the multiplier of the number of added edges within each partition relative to the original graph. Using this expanded edge set, a second GNN is trained to produce

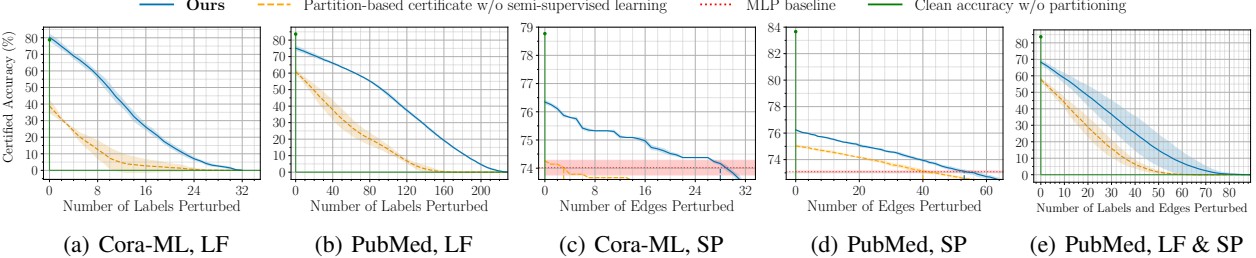

| (a) Cora-ML, LF | (b) PubMed, LF | (c) Cora-ML, SP | (d) PubMed, SP | (e) PubMed, LF & SP |

*Figure 4.* The effectiveness of our method is demonstrated by the increase in certified accuracy compared to the naive partition-based approach from Sec. 4. (a) and (b) are certified against Label Flipping (LF) with $k = 80$ and $k = 500$ partitions, respectively; (c) and (d) are certified against Structure Perturbations (SP) with $k = 800$ partitions; (e) is certified against both (LF&SP) with $k = 250$ partitions. We showcase the stark improvement of robustness both on Cora-ML and PubMed, demonstrating that our method works for both smaller and larger graphs. The red lines in (c) and (d) represent the performance of an infinitely robust MLP, serving as a trivial baseline for structure perturbation. The green dots are the non-robust clean accuracies of the same model trained without any partitions.

final predictions used for certification. Our certification pipeline against structure perturbation is outlined in Algo. 2. On first sight, a downside of our link prediction scheme is that we sample a denser graph for each partition than the original one. If this is implemented naively, it will have roughly $O(kn^2)$ time and memory complexity, which does not scale well with the graph size and the number of partitions. To address this, we introduce an efficient algorithm in App. D.1, which has a near-linear complexity based on managing a global edge-candidate heap.

With the proposed methods addressing either label flipping or structure perturbations, a scheme for certifying against *both types of attacks* becomes possible. We partition both labels and edges into partitions, perform semi-supervised edge and label generation iteratively, and obtain the final base classifier by training on the extended edges and labels, before we take the majority vote and compute certificates. We provide pseudocode for this joint pipeline in App. D.2.

## 6. Experimental Evaluation

In this section, we investigate the robustness guarantees derived by Deep Self-Training Graph Partition Aggregation (ST-GPA) and showcase the improvement of ST-GPA compared to simple partition aggregation on graphs.[2]

**Experimental details.** We demonstrate results for transductive node classification on four datasets: Cora-ML (Bojchevski & Günnemann, 2018), and the three Planetoid datasets CiteSeer, Cora, and PubMed (Yang et al., 2016); and for three GNNs: Graph Convolutional Networks (GCN) (Kipf & Welling, 2017), Graph Attention Networks (GAT) (Veličković et al., 2018), and APPNP (Klicpera et al., 2019). To train a robust classifier, we partition graphs as described in Sec. 5. Each round of semi-supervised learning adds pseudo-labels or edges to the training sets of the individual

partitions, while keeping the partitions isolated. We train an ensemble classifier after each round of pseudo-label or pseudo-edge generation to investigate the effect of the individual semi-supervised learning steps. Results are reported using *certified accuracy*, which is the percentage of test nodes whose predictions are correct and provably robust, as a function of the perturbation size defined in Eq. (1). In all figures, colored areas represent the standard deviation over 3 seeds. We represent a baseline ensemble classifier trained on partitions without any semi-supervised learning as dashed lines. It is important to note that MLPs exhibit infinite robustness against structure perturbations, since they do not utilize edges during training. Consequently, any model with certified accuracy against structure perturbations below that of an MLP is trivial. Thus, we include an MLP as baseline as a red-dotted line in the structure perturbation plots. In these plots, we discard the trivial part of the curve below this baseline. We provide further details on our setup in App. A.

**ST-GPA yields strong certified robustness.** Fig. 4 demonstrates the stark improvement of certified accuracies by our proposed certification method ST-GPA, against Label Flipping (LF) in Figs. 4(a) and 4(b), Structure Perturbation (SP) in Figs. 4(c) and 4(d), and both label and structure perturbations in Fig. 4(e). If partitioning is applied naively, i.e., without our semi-supervised learning schemes from Sec. 5, we only get marginal certified accuracy curves (orange dashed lines) due to the sparsity of labels or edges as discussed in Sec. 4. In the structure case on Cora-ML (Fig. 4(c)), the clean accuracy of an ensemble GCN is even worse than an infinitely robust MLP, rendering the naive approach ineffective. In contrast, with our method we restore the ensemble classifier's clean accuracy to a higher level compared to an ensemble without semi-supervised learning, typically around 70% to 80%, and this also allows the certified accuracy curves to drop down slower, meaning higher certified accuracy against the same number of perturbations. We include as a reference the clean accuracy of a GCN trained without any partitioning, and thus leading

---

[2]The code to reproduce our results can be found at: https://github.com/p-CXTN/ST-GPA.

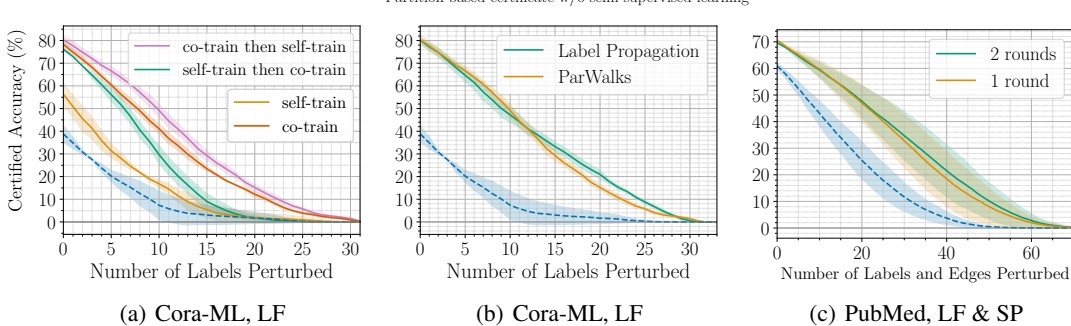

*Figure 5.* (a) Different orders of co-training and self-training against Label Flipping (LF) on Cora-ML with $k = 80$ and label propagation as co-training method; (b) Label propagation provides similar performance in co-training compared to ParWalks (Wu et al., 2012) while being scalable to larger graphs, demonstrated on Cora-ML with 80 partitions; (c) Our method provides significant improvements against Label Flipping (LF) and Structure Perturbations (SP) on PubMed with 200 partitions with link prediction, co-training then self-training, while stacking more rounds of semi-supervised learning in said order yields further but marginal improvement.

to certified accuracies of 0% for any non-zero perturbation. Table 2 in App. A reports clean accuracies on other datasets.

For certifying label flipping, we demonstrate the effect of the order of co- and self-training in Fig. 5(a). We find that first co- and then self-training generally works best. This can be understood as self-training involves training a GNN on the available labels in a partition. With $k = 80$ partitions, each partition contains only ~10 labels on smaller datasets. In contrast, co-training with label propagation does not require a representation learning step and effectively leverages the graph structure despite the extremely low label rate, producing high-quality pseudo-labels that can be further improved by subsequent self-training rounds. This effect is consistent across all investigated datasets as we show in Fig. 8 in App. B.2. Consequently, we adopt this order for all experiments involving label flipping.

In our certificate against both label flipping and structure perturbations, we perform link prediction, then co-training with label propagation, and finally self-training on labels. We perform several rounds of edge and label generation in this order because label propagation relies on the meaningful graph structure that link prediction generates, and self-training generally works better after co-training, as we find out in Fig. 5(a). In Fig. 5(c) we show that our method introduces significant improvements in the first round, where additional rounds can lead to further improvements.

**Co-training with label propagation is similarly performant as ParWalks yet scalable.** Since Li et al. (2018) proposed ParWalks in combination with self-training to address low label rates, we compare ParWalks with the Label Propagation (LP) from Zhu & Ghahramani (2003). As shown in Fig. 5(b), the performance advantage of ParWalks over LP is negligible. This is consistent across different datasets as shown in Fig. 10 in App. B.2. Given that LP with random teleportation does not require computing the inverse of the Laplacian, we adopt LP as our primary method.

**Our method works with any GNN.** The first row in Fig. 6 shows the certification performance of a GCN, GAT, and APPNP with and without our approach. The results show similar performance improvements and trends across all models, with clean accuracy boosted to around 80% for LF, 76% for SP, and 67% for both. We highlight this because our partition-based certificate is architecture-agnostic, and does not assume anything about the underlying classifiers themselves. Thus, our method stays compatible with better underlying GNNs, for which one can expect to produce better certificates. Exemplary, in App. B.4 we use a modern graph transformer as base classifier (Wu et al., 2022).

**Our method scales well with partitions.** The second row in Fig. 6 shows the effect of varying the number of partitions $k$. The results demonstrate the effectiveness and scalability of our method: as $k$ increases, which is necessary to derive stronger guarantees, the baseline performance rapidly declines due to the sparsity of labels and structure per partition. However, our experiments indicate that link prediction, co-training, and self-training are required for non-trivial robustness guarantees. This is supported by more results in Fig. 11 in App. B.2. We observe that dividing labels to more than 100 partitions becomes impractical given the training size of 30% on Cora-ML, as partitioning would result in some partitions containing no labeled nodes, making training on those partitions infeasible. Further experiments on the selectiveness hyperparameter $t$ and $\varepsilon$, and label propagation teleportation parameter $\alpha$ are included in App. B.5.

**Our label flipping certificate scales well to large graphs.** In App. B.3 we show that our label flipping certificate scales well to large graphs such as ogbn-arxiv, or significantly more dense graphs such as Wiki-CS. For more details on the time complexities and scaling behavior of both our label flipping and structure certificate, we refer to App. F, as well as to our discussion in Sec. 8.

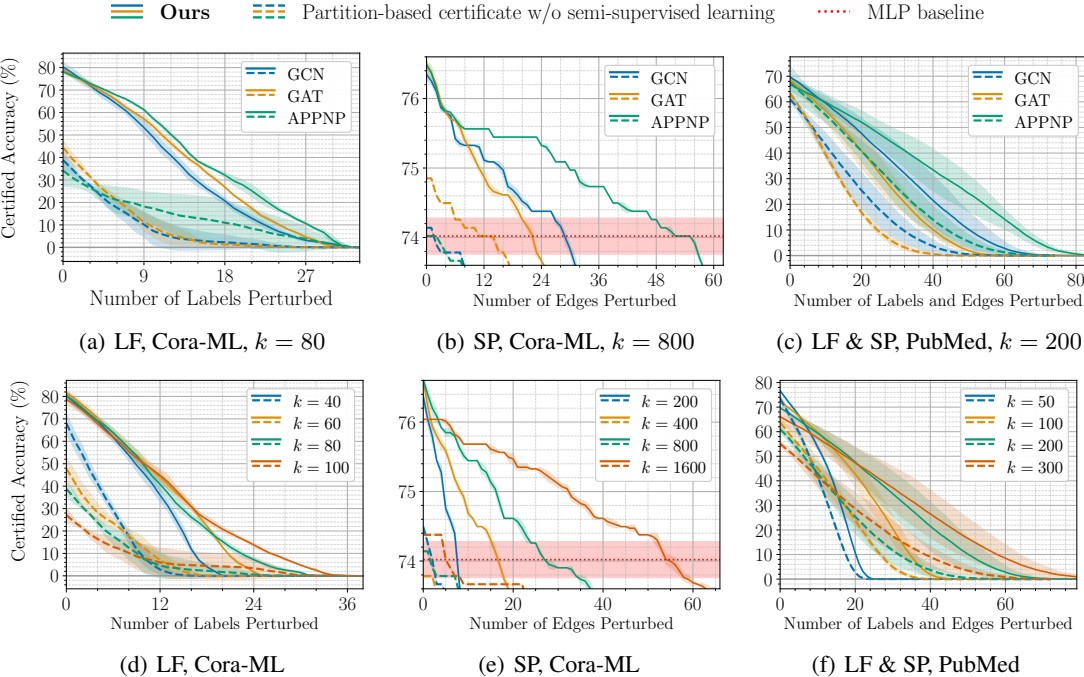

*Figure 6.* In (a) to (c) our method improves upon naive partition-based certification w/o semi-supervised learning, regardless of the GNN type; (d) to (f) demonstrate for GCNs that our method scales very well for larger number of partitions $k$, which provides better certificates.

## 7. Related Work

Studying upper bounds on adversarial robustness against test- and training-time perturbations through adversarial attacks is well established for message-passing GNNs (Dai et al., 2018; Zügner et al., 2018; Zügner & Günnemann, 2019; Mujkanovic et al., 2022; Günnemann, 2022), as well as for more modern graph transformers (Foth et al., 2025). Self-training has been successfully used in empirical defenses against such attacks (Li et al., 2024; Lee & Park, 2025), but we are not aware of an application of self-training to improve certified robustness. In contrast to adversarial attacks, robustness certificates provide a lower bound on adversarial robustness, and certificates against test-time attacks are well researched for i.i.d. data (Li et al., 2023), as well as for the graph domain (Günnemann, 2022; Scholten et al., 2022; 2023; Hojny et al., 2024). However, there are few works studying certification against changes to the training data. For the image domain, there are three main approaches: (i) partition-and-aggregate (Levine & Feizi, 2021), (ii) randomized smoothing (over the training data) (Weber et al., 2023), and (iii) differential privacy (Ma et al., 2019); and we refer to Gosch et al. (2025) for a representative survey.

Most related to our work is the partition-and-aggregate scheme by Levine & Feizi (2021), with various follow-up works (Wang et al., 2022; Chen et al., 2022; Rezaei et al., 2023; Scholten & Günnemann, 2025; Mohgaonkar et al., 2026; Saxena et al., 2026). However, it was mainly studied

in the image domain so far. Regarding graphs, Lai et al. (2024b) develop a collective, probabilistic poisoning certificate against node injection based on randomized smoothing (Lai et al., 2024a), which however, is not applicable to the perturbation models studied in this work. Gosch et al. (2025) develop a certification paradigm to certify node-feature poisoning, extended to label poisoning by Sabanayagam et al. (2025), but both cases are limited to infinite-width GNNs. Further, the label certificate by Sabanayagam et al. (2025) does only scale to graph datasets having at most 100-200 training labels and thus, does not scale to even our smallest datasets. However, we still provide a comparison to Sabanayagam et al. (2025) in App. E.1. Concurrently to our work, Mohgaonkar et al. (2026) extended the scalability of the label certificate by Sabanayagam et al. (2025) to scale to image datasets. However, the scaling is only possible at additional costs of the clean accuracy, and again, only valid for infinite-width networks. Further, they did not study graph certification. Li et al. (2025) apply partitioning to derive poisoning certificates for GNNs. However, they do not certify against label flipping, and their structure certificates are below the performance of an MLP (see App. E.2) and thus, vacuous. Further, their feature certification is only applicable to graph classification, as every node is in all partitions. Overall, our work is the first to provide effective, non-vacuous certified robustness guarantees for GNNs against both label and structure poisoning in node classification tasks.

## 8. Discussion and Limitations

We've shown that our semi-supervised training schemes are essential for meaningful robustness guarantees in the graph domain with partition-based methods. In this section, we address and discuss the limitations of our method and show how this hints to promising research directions.

**Application to heterophilic graphs.** While our certification framework is general, the chosen semi-supervised training components rely fundamentally on homophily, which limits their applicability to heterophilic graphs, which we empirically quantify through a study on synthetic graphs in App. H. Thus, addressing heterophily requires different semi-supervised learning approaches, such as novel label propagation or link prediction methods tailored to heterophilic structures, each of which represents its own independent research challenge. However, the core ideas of partitioning and self-training remain applicable and could inspire effective robustness guarantees if adapted properly to heterophilic settings.

**Link prediction can be expensive.** Furthermore, while the label certificate scales well to larger graphs, another important limitation is the computational overhead of link prediction on denser or larger graphs, necessary for effective structure certificates. Our method can produce semi-dense partitions by adding edges to capture the graph structure and diluting the effect of noisy inter-class edges, increasing memory and runtime costs. This overhead prevents scaling link prediction effectively to large datasets like ogbn-arxiv, as shown in our scalability analysis in App. F. While alternative link prediction algorithms might alleviate this, most are designed for full graphs rather than sparse partitions, making this a non-trivial challenge.

For a detailed discussion on *tightness*, we refer to App. G.

## 9. Conclusion

We derive a self-training framework (ST-GPA) that significantly improves certified robustness against poisoning in sparse graph-structured domains. By adding both synthetic labels and structure through effective semi-supervised learning techniques, our method overcomes the limitations of existing partition-based approaches. Empirical results show large improvements in certified robustness to both label and structure poisoning without compromising clean accuracy. Our findings highlight that effectively leveraging semi-supervised learning on sparse data is essential for provably robust graph machine learning against poisoning through partition-based approaches. This potentially offers a promising direction for building more robust models in semi-supervised settings beyond the graph domain.

## Impact statement

This work advances the field of robustness certification by enabling robustness of graph neural networks against structure and label poisoning attacks, thereby fostering more reliable and trustworthy machine learning. While there might be many further potential societal consequences of our work, none which we feel must be specifically highlighted here.

## Acknowledgements

This work has been funded by the DAAD program Konrad Zuse Schools of Excellence in Artificial Intelligence (sponsored by the Federal Ministry of Education and Research) and by the German Research Foundation, grant GU 1409/4-1. The authors of this work take full responsibility for its content.

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

# A. Experiment Setup

**Datasets.** As described in the beginning of Sec. 6, we use the 3 Planetoid datasets (Yang et al., 2016) Cora, CiteSeer and PubMed, available on PyTorch Geometric,[3] and the citation dataset Cora-ML (Bojchevski & Günnemann, 2018). As a standard procedure in graph machine learning (Shchur et al., 2018), we preprocess all the datasets by taking the largest connected component and force the graph to be undirected. The statistics of the dataset we use can be found in Table 1. The training, validation and test set nodes are determined with scikit-learn's train_test_split() function,[4] with random_state fixed to 12138. We first separate the test nodes' indices from training and validation (and unused), then use this function again to separate the training set from the validation with the same seed. Only the 30% training labels are available to the model during training. Additionally, we evaluate the label certificates on Wiki-CS (Mernyei & Cangea, 2020) and ogbn-arxiv (Hu et al., 2020), and we sample 1/10th of their original training labels. The 2 Amazon datasets (Shchur et al., 2018), Amazon-Photo and Amazon-Computers, are also used for evaluating label certificates, and the same scheme as the Planetoid datasets for splitting the nodes is used.

*Table 1.* Statistics of Datasets. The number of training labels is 30% of all nodes for Cora-ML, CiteSeer, Cora, PubMed; 60% for Amazon-Computers and Amazon-Photo; and 5% for Wiki-CS and ogbn-arxiv.

| Name | # Nodes | # Training Labels | # Edges | # Features | # Classes | Avg. Degree |
|---|---|---|---|---|---|---|
| Cora-ML | 2810 | 843 | 7981 | 2879 | 7 | 2.84 |
| CiteSeer | 2110 | 633 | 3668 | 3703 | 6 | 1.74 |
| Cora | 2708 | 812 | 5069 | 1433 | 7 | 1.87 |
| PubMed | 19717 | 5915 | 44324 | 500 | 3 | 2.25 |
| Wiki-CS | 11701 | 586 | 216123 | 300 | 10 | 18.47 |
| Amazon-Computers | 13752 | 8251 | 491722 | 767 | 10 | 35.76 |
| Amazon-Photo | 7650 | 4590 | 238162 | 745 | 8 | 31.13 |
| ogbn-arxiv | 169343 | 9600 | 1166243 | 128 | 40 | 6.89 |

*Table 2.* Model Clean Accuracies on Datasets. We report the clean accuracy (percentage) on the test nodes with given dataset and model with the standard deviation of 3 repeated experiments with different initialization seeds to the model.

| | MLP | GCN | GAT | APPNP |
|---|---|---|---|---|
| Cora-ML | 74.02(0.26) | 78.77(2.10) | 76.95(0.37) | 85.13(0.24) |
| CiteSeer | 70.35(0.20) | 67.46(0.45) | 64.98(2.38) | 72.30(0.61) |
| Cora | 67.78(0.06) | 76.18(0.41) | 75.77(1.22) | 83.39(0.82) |
| PubMed | 73.09(0.05) | 83.66(1.33) | 79.01(0.56) | 87.49(0.41) |

We follow a 30%-10%-30% train-validation-test split of node labels for all experiments except for larger datasets following Li et al. (2025). Note that the 10% validation set is typically used for regularization tasks such as hyperparameter tuning and early stopping when training a single GNN. However, since we use a fixed set of hyperparameters and do not employ early stopping, the validation set is not utilized during training.

**Model Parameters.** We used Graph Convolutional Networks (GCN)(Kipf & Welling, 2017), Graph Attention Networks (GAT), and Approximate Personalized Propagation of Neural Predictions (APPNP)(Klicpera et al., 2019) throughout our evaluations.

Our GCNs consist of 2 layers of GCNConv layer with added self-loops from pytorch geometric.[5] At the bottleneck we use 8 hidden channels, a dropout layer, and ReLU activation. For GAT we use 2 GATConv layers from pytorch geometric.[6] The first layer condense the feature channels to 8 hidden channels with 8 attention heads, and applies dropout and ELU activation, and the second layer then condense the $8 \times 8$ dimension in the middle to class-wise logits. The APPNP model consists of a 2 layer MLP with 8 hidden channels and then an APPNP layer from pytorch geometric[7] to propagate representations,

---

[3] https://pytorch-geometric.readthedocs.io/en/2.6.0/modules/datasets.html

[4] https://scikit-learn.org/stable/modules/generated/sklearn.model_selection.train_test_split.html

[5] https://pytorch-geometric.readthedocs.io/en/2.5.2/generated/torch_geometric.nn.conv.GCNConv.html

[6] https://pytorch-geometric.readthedocs.io/en/latest/generated/torch_geometric.nn.conv.GATConv.html

[7] https://pytorch-geometric.readthedocs.io/en/latest/generated/torch_geometric.nn.conv.APPNP.html

with number of iterations $K = 10$, and teleport probability $\alpha = 0.1$. In structure perturbation experiments specifically, we don't use dropout as it provides us with better results. For additional evaluation of label certificates on a modern graph transformer, we use NodeFormer (Wu et al., 2022) and the exact setup as their published code. Additionally, we use an 2 layered MLP as baseline against structure perturbations. It also has 8 hidden units and no dropouts. Note that here in our setup, the GCN with an empty graph is strictly equivalent to the MLP. Both other attack models use dropouts with a probability of 0.5. All other unstated parameters follow the pytorch geometric default.

**Hyperparameters for Training.** For GNN training, we use a fixed set of hyperparameters inherited from Li et al. (2018) , which are commonly used in GNN models. Specifically, all models are trained for 200 epochs without early stopping, using the Adam optimizer with a learning rate of 0.01 and a weight decay of $5 \times 10^{-4}$. However, due to the unstable nature of GNN training, we choose the epoch with the lowest loss for prediction. To calculate this without overfitting, we separate the data in each partition in halfs, and use one half for training and another for evaluation of the loss. Note that this split happens to the partitioned 30% labels or edges each partition have access to.

For label propagation, we iterate until the score matrix converges, but cap off at 100 iterations.

**Evaluation Process.** As described in Sec. 5, either all edges, the 30% training set labels, or both are partitioned for the weak classifiers, depending on different types of attacks. We perform transductive node classification, meaning that the weak classifiers then have access to all the graph data $G = (\mathcal{E}, \boldsymbol{X}, \boldsymbol{y})$ except for the poisoned items, which are partitioned in the first place.

All models are evaluated with at least 3 different deterministically chosen seeds for model initialization to evaluate the repeatability of our results. The seeds are chosen by python's random module. We seed the random module with a fixed seed of 123456, and then use random.randint(0, 2**32) to generate a deterministic random seed each time we repeat an experiment.

After the training is done, we evaluate the certified accuracies on the 30% test nodes. This is done by storing the class predictions of each weak classifier for each node and calculate the robust margin as introduced in Eq. (8). Then the certified accuracies we reported are given by calculating the ratio of test nodes whose robust margin, together with a specific perturbation size, satisfies Eq. (8).

**Hyperparameters Summary.** Here we provide a table of all hyperparameters involved as a summary to our description of the experiment setup, alongside their default values and selection criteria. Throughout this paper, all hyperparameters take their default values (if any) unless otherwise stated.

*Table 3.* Hyperparameters

| hyperparameter | description | attack models | related models | default value | selection criteria |
|---|---|---|---|---|---|
| lr | learning rate | all | all | 0.01 | follows Li et al. (2018) |
| wd | weight decay | all | all | $5 \times 10^{-4}$ | follows Li et al. (2018) |
| ep | training epochs | all | all | 200 | follows Li et al. (2018) |
| es | early stopping | all | all | None | follows Li et al. (2018) |
| init_seed | seed for python random module | all | all | 123456 | / |
| repetition | number of repeated experiments | all | all | 3 | / |
| num_layers | number of layers in the model | all | all | 2 | prevents overfitting |
| hidden_size | hidden channels | all | all | 8 | prevents overfitting |
| dropout | dropout probability in dropout layers | all | GCN, GAT | 0.5 | prevents overfitting |
| activation | activation function between 2 layers | all | GCN, APPNP, MLP | ReLU | / |
| | | all | GAT | ELU | / |
| train_size | % of labels for training | all | all | 30% | / |
| val_size | % of labels for validation | all | all | 10% | / |
| test_size | % of labels for testing | all | all | 30% | / |
| k | number of partitions | LF | all | 80 | Fig. 6 |
| | | SP | | 800 | Fig. 11 |
| | | LF&SP | | 200 | Fig. 6 |
| order | order of co-training and self-training | LF, LF&SP | all | C/T then S/T | Fig. 8 |
| co-train method | ParWalks(PW) or label propagation(LP) | LF, LF&SP | all | LP | Fig. 10 |
| t | number of pseudo-labels per class | LF, LF&SP | all | 50 | Fig. 14(a) |
| $\alpha$ | teleport probability in label propagation | LF, LF&SP | all | 0.9 | Fig. 14(b) |
| $\varepsilon$ | $\times \varepsilon$ pseudo-edges than original graph | SP | all | dataset specific | Fig. 15 |

# B. Additional Experiment Results

## B.1. Clean Accuracy of Ensembles

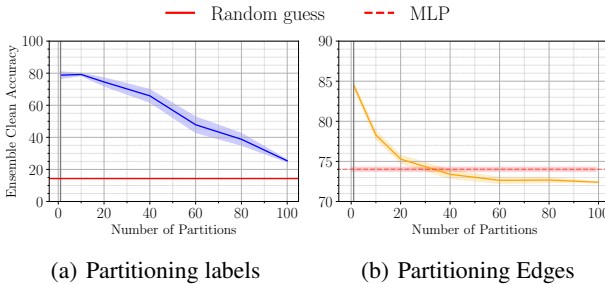

(a) Partitioning labels      (b) Partitioning Edges

*Figure 7.* Deterioration of clean accuracy of ensembles on Cora-ML. Standard deviations are reported as colored areas over 3 different seeds.

In Fig. 7 we show the clean accuracies of an ensemble classifier if the partitioning scheme is applied trivially, i.e. without semi-supervised learning. In the label case, the sparsity degrades the clean accuracy of the ensemble as $k$ goes larger. With $k = 100$ partitions, the clean accuracy drops to just over 20%. In the edge partitioning case, lack of edge information degrades the clean accuracy of an ensemble below that of an MLP's over about $k = 30$ partitions. The deterioration of performance shown here necessitate the introduction of semi-supervised learning schemes within the sparse partitions.

## B.2. Additional Results on Other Datasets

Here we present results on other datasets supporting our conclusion from Sec. 6, which have been presented in the main part based on one exemplary dataset. Note that the subfigures marked with (*) are already present in Sec. 6.

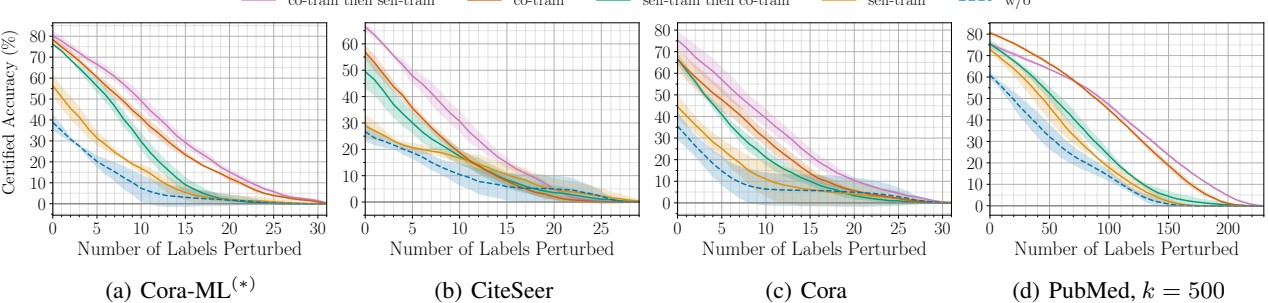

(a) Cora-ML[(*)]     (b) CiteSeer     (c) Cora     (d) PubMed, $k = 500$

*Figure 8.* We show different orders of co-training and self-training against label flipping on different datasets. $k = 80$ unless otherwise stated.

**The order of co-train then self-train generally works the best against label flipping.** Fig. 8 showcases this. Our method restores the clean accuracy to generally 70% to 80%, while having higher certified accuracies, despite the poor performance with vanilla partitions.

**Against both label and structure perturbations on smaller datasets, our method provides improvement but the limit of $k$ prevents meaningful joint certificates.** Due to the extreme sparsity of labels and edges if we are facing poisoning of both, we can't use too many partitions, typically limiting $k$ to around 10. Although the curve drops too fast due to a small $k$, yielding hardly any usable certified accuracy against even 1 poisoned label or edge, our method still provides 15% to 20% clean accuracy increase.

**Co-training with label propagation is similarly performant as ParWalks yet scalable.** Our method has similar certified accuracy compared to ParWalks across datasets, except for CiteSeer where it's about 10% less. However, we argue that this is compensated by the vastly shorter training time and the characteristic that scales easily to larger datasets by our method.

**Link prediction allows arbitrarily large $k$ which generates even better certificates.** In Fig. 11, we demonstrate the near-perfect scalability of link prediction on generating certificates. The number of partitions $k$ determines the maximum achievable robustness with the partitioning scheme. Therefore, given sufficient computational resources, increasing $k$

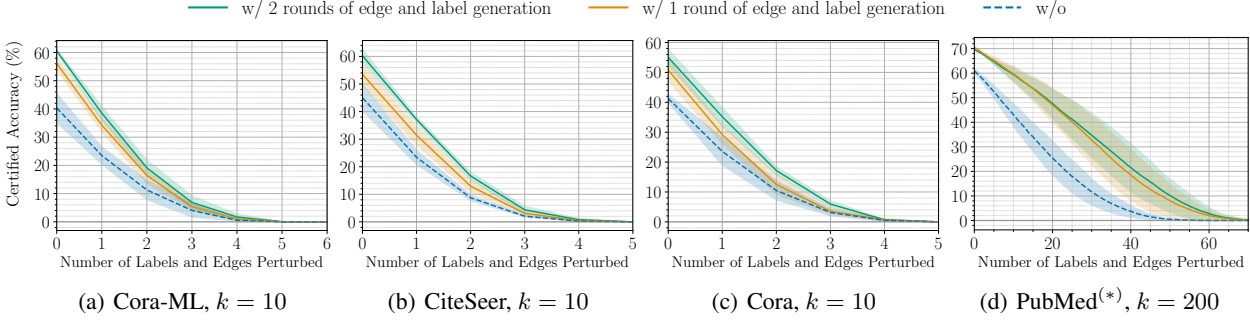

*Figure 9.* Different number of rounds of link prediction + co-training + self-training against both label and structure perturbations.

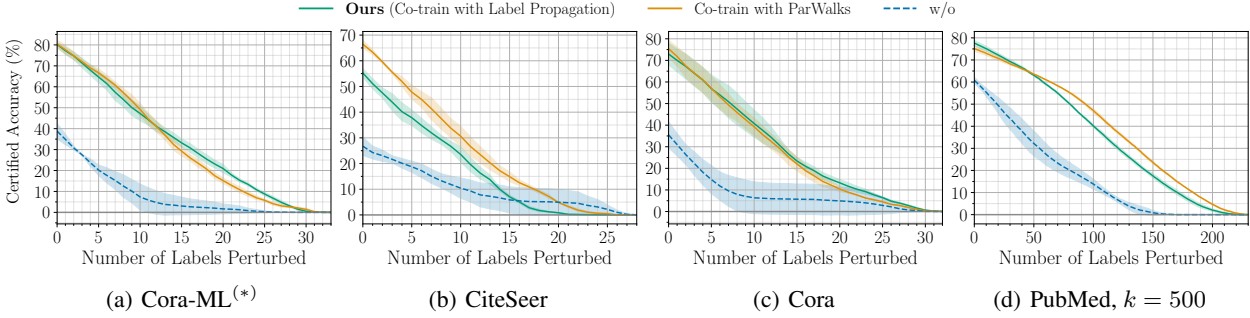

*Figure 10.* Comparison between our method (co-training with label propagation) and ParWalks(Wu et al., 2012). The results are achieved by co-training then self-training on given datasets, with $k = 80$ partitions unless otherwise stated.

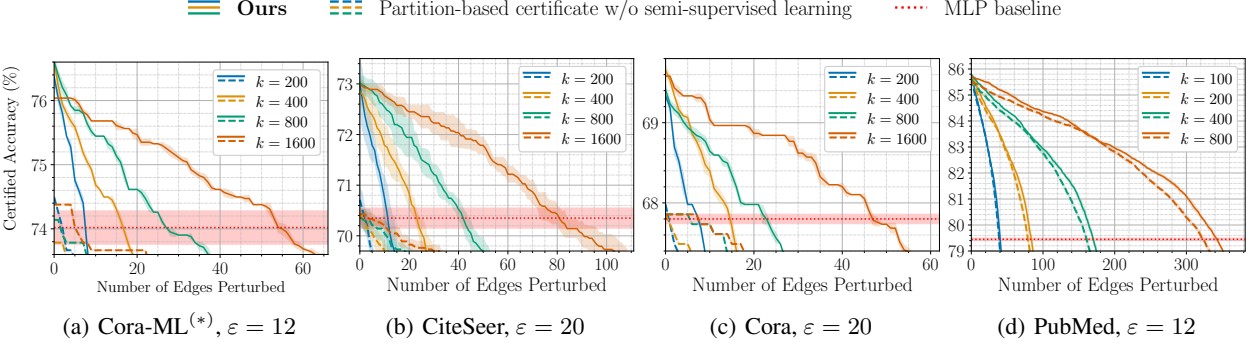

*Figure 11.* Scalability over $k$ with our link prediction method.

directly leads to improved robustness. Furthermore, we emphasize that $k$ is theoretically unbounded in its capacity to enhance robustness against structure perturbations.

### B.3. Certifying Label Flipping on Larger and More Diverse Datasets

**Our co-training and self-training scheme also scales to larger and more diverse datasets.** Fig. 12 shows the performance gain upon a trivial partitioning scheme on Wiki-CS, amazon-computers, amazon-photos, and ogbn-arxiv. Wiki-CS and the amazon datasets include significantly more edges than the citation datasets. ogbn-arxiv includes significantly more nodes and edges (see also Table 1 in App. A). Our co-training and self-training generally restores clean accuracy well, while also having a higher certified accuracy. For Wiki-CS and ogbn-arxiv we used 1/10 of the label rate as other datasets to demonstrate this improvement with fewer partitions. Using the original split, a larger $k$ would also be possible, and likely improve results (compare Figs. 12(d) and 12(e)). Our structure certificate struggles to scale well to such large and dense datasets, as we discuss in Sec. 8.

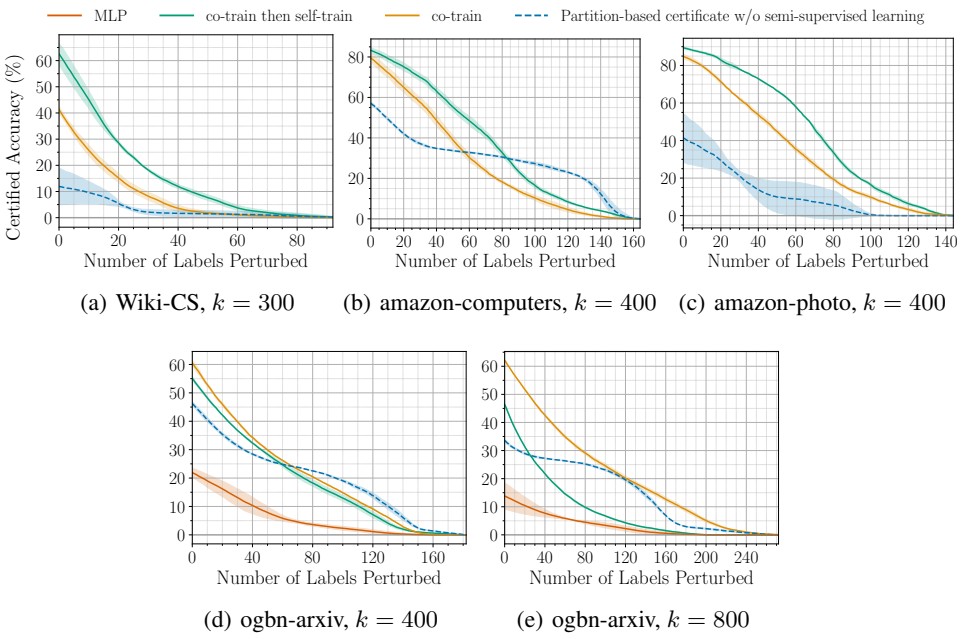

*Figure 12.* Certifying label flipping on larger and denser graph datasets. MLP baseline for ogbn-arxiv refers to an MLP ensemble.

## B.4. Additional Results for Graph Transformers

**Our co-training and self-training scheme also applies to modern graph transformers.** Fig. 13 shows that our method improves certified accuracy from 20% to over 70%, showcasing that it works well with modern graph transformer architectures, apart from the result in Figs. 6(a) and 6(b) in Sec. 6. In particular, we find that NodeFormer w/o our semi-supervised learning strategies struggles to learn from sparsely labeled partitions. However, enriching the partitions with pseudo-labels using our proposed co-training scheme leads to good clean accuracies and certified robustness of the GT-ensemble against label poisoning.

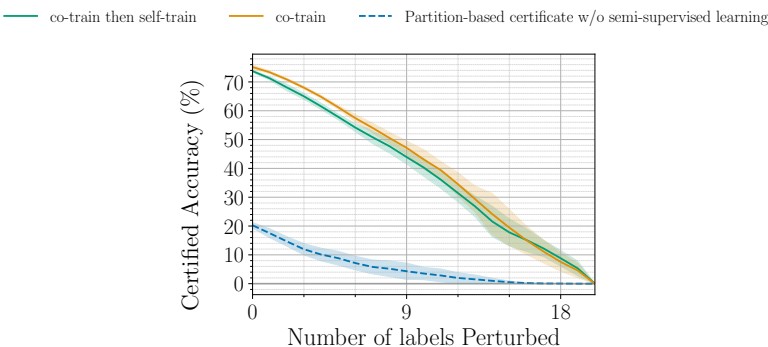

*Figure 13.* Certifying label flipping on Cora-ML using NodeFormer ($k = 40$).

## B.5. Ablations on Hyperparameters

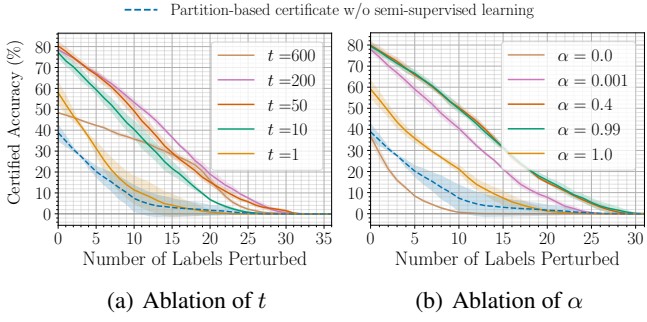

*Figure 14.* Certifying label flipping on Cora-ML with $k = 80$ partitions.

**The hyperparameter $t$ offers a way to balance between performance and robustness.** Fig. 14(a) illustrates the impact of varying the number of pseudo-labels $t$ added during training. When $t$ is small, such as $t = 1$ or $t = 10$, the pseudo-labels are too selective and insufficient in quantity to effectively train the subsequent GNN. Conversely, when $t$ is too large, for example $t \geq 600$, the quality of the pseudo-labels deteriorates, which negatively affects the ensemble classifier's performance. Therefore, selecting an appropriate range for $t$ is critical to achieving optimal certified accuracy. For Cora-ML, this corresponds roughly to the range $50 \leq t \leq 200$. Since $t = 50$ works already quite well and introduce less computation overhead, we fix $t = 50$ for all experiments.

**The random teleport probability $\alpha$ is pretty robust, but need to be properly chosen.** Fig. 14(b) shows the certificates with co-training then self-training, but varying the random teleport probability $\alpha$ An $\alpha = 0$ means always randomly teleport to a labeled node in co-training, while an $\alpha = 1$ means no random teleport. As demonstrated by the results, the performance generally plateaus when $0.01 \leq \alpha \leq 0.99$. As long as $\alpha$ isn't chosen to be too large or too small, the certified accuracy remains the best achievable one. We choose to fix $\alpha = 0.9$ for all experiments.

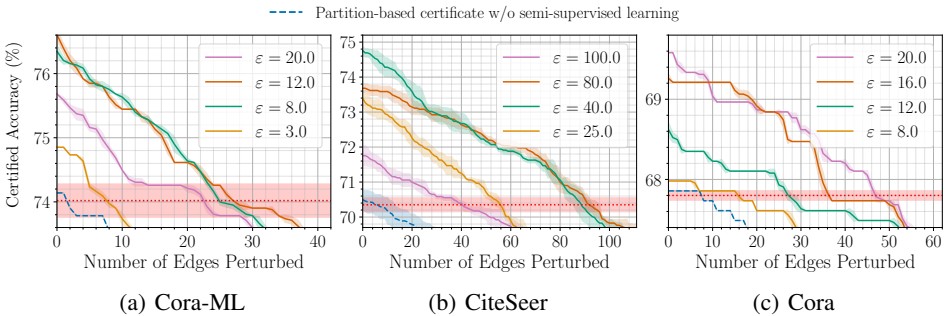

*Figure 15.* Certifying structure perturbations on the three smaller datasets with $k = 800$ partitions.

**There is a best $\varepsilon$ for each dataset that we can tune to maximize robustness.** $\varepsilon$ serves as a control parameter for the selectiveness in adding pseudo-edges. A too small $\varepsilon$ fails to capture sufficient graph structure, while a too large $\varepsilon$ introduces noise that reduces the signal-to-noise ratio in the generated graph, thereby degrading performance. So there is a best $\varepsilon$ in between, and according to Fig. 15, this best value varies a lot between datasets, even though the size of which are similar. To avoid high overheads in all other experiments, we cap $\varepsilon$ to 20.

## C. Proof of Theorems

### C.1. Proof of Generalized DPA

We restate Thm. 3.1 before proving it:

**Theorem 3.1** (Generalized DPA). *Given a clean, possible non-i.i.d and structured dataset $D$, and a poisoned dataset $\tilde{D}$, the majority-vote classifier prediction remains unchanged, i.e., $g_D(x) = g_{\tilde{D}}(x)$, as long as $d_h d(D, \tilde{D}) \leq \lfloor (n_c(D, x) - \max_{c' \neq c} (n_{c'}(D, x) + \mathbb{1}_{c' < c})) / 2 \rfloor = r_m(D, x)$, where $c = g_D(x)$ is the predicted class on the clean dataset.*

*Proof.* We first introduce a lemma that gets rid of the floor operation:

**Lemma C.1** (Floor operation equivalency)**.**

$$a \leq \lfloor \frac{b}{2} \rfloor \Leftrightarrow 2a \leq b, \forall a, b \in \mathbb{N} \tag{10}$$

*Proof.* $a \leq \lfloor \frac{b}{2} \rfloor \leq \frac{b}{2} \Rightarrow a \leq \frac{b}{2}$ and $2a \leq b \Rightarrow a \leq \frac{b}{2} \Rightarrow a = \lfloor a \rfloor \leq \lfloor \frac{b}{2} \rfloor$  □

We use the shorthand notation of $n_c$ being $n_c(D, x)$ and $\tilde{n}_c$ being $n_c(\tilde{D}, x)$, and $r = d(D, \tilde{D})$.

Given Thm. 3.1 and Lemma C.1

$$2d_h \cdot r \leq n_c - \max_{c' \neq c}(n_{c'} + \mathbf{1}_{c' < c}) \tag{11}$$

We get rid of the max operation by taking any class and move all terms to one side

$$0 \leq n_c - n_{c'} - \mathbf{1}_{c' < c} - 2d_h \cdot r, \forall c' \neq c \tag{12}$$

Because the training is conducted in a deterministic manner, the classifier will give the same prediction if the training data is the same. So the number of weak classifiers that predicts differently for a node $f_i(v) \neq \tilde{f}_i(v)$ is also at most $d_h \cdot r$. So for any class, the number of predictions changed is bounded by

$$\forall \bar{c} \in [C], |n_{\bar{c}} - \tilde{n}_{\bar{c}}| \leq d_h \cdot r \tag{13}$$

From Eq. (13) we plug in $\bar{c} = c$ and $\bar{c} = c'$ to get

$$n_c - \tilde{n}_c \leq d_h \cdot r \tag{14}$$
$$\tilde{n}_{c'} - n_{c'} \leq d_h \cdot r \tag{15}$$

which is equivalent to

$$n_c \leq \tilde{n}_c + d_h \cdot r \tag{16}$$
$$-n_{c'} \leq -\tilde{n}_{c'} + d_h \cdot r \tag{17}$$

Plugging this to Eq. (12), we have

$$\forall c' \neq c, 0 \leq n_c - n_{c'} - \mathbf{1}_{c' < c} - 2d_h \cdot r \tag{18}$$
$$\leq \underbrace{\tilde{n}_c + d_h \cdot r}_{\text{Eq. (16)}} \underbrace{-\tilde{n}_{c'} + d_h \cdot r}_{\text{Eq. (17)}} - \mathbf{1}_{c' < c} - 2d_h \cdot r \tag{19}$$
$$= \tilde{n}_c - \tilde{n}_{c'} - \mathbf{1}_{c' < c} \tag{20}$$

So for the ensemble classifier trained on poisoned data

$$\tilde{n}_c \geq \max_{c' \neq c}(\tilde{n}_{c'} + \mathbf{1}_{c' < c}) \tag{21}$$

and the ensemble classifier's prediction is unchanged because $c$ is the majority class.  □

### C.2. Proof of Distance Function

The following is a proof that $d((\mathcal{E}, \boldsymbol{y}), (\tilde{\mathcal{E}}, \tilde{\boldsymbol{y}}))$ is a distance.

*Proof.* We recall that $d((\mathcal{E}, \boldsymbol{y}), (\tilde{\mathcal{E}}, \tilde{\boldsymbol{y}}))$ is defined to be the sum of symmetric set difference between edges and the hamming distance between labels

$$d((\mathcal{E}, \boldsymbol{y}), (\tilde{\mathcal{E}}, \tilde{\boldsymbol{y}})) = \Delta(\mathcal{E}, \tilde{\mathcal{E}}) + \delta(\boldsymbol{y}, \tilde{\boldsymbol{y}}) \tag{22}$$

and a function $d : M \times M \to \mathbb{R}$ is a distance on metric space $M$ if $\forall x \in M$, $d(x, x) = 0$, $d(x, y) > 0, x \neq y$, $d(x, y) = d(y, x)$, and $d(x, y) \leq d(x, z) + d(y, z)$.

We first note that $d(\cdot, \cdot)$ between one $(\mathcal{E}, \boldsymbol{y})$ and itself is $0$ because both its terms are $0$; It's value is always positive as it's the sum of two positive numbers, if both input are distinct; $d(\cdot, \cdot)$ is symmetric because both its terms are symmetric and changing the order of objects doesn't affect the value.

To prove triangle inequality

$$d((\mathcal{E}_1, \boldsymbol{y}_1), (\mathcal{E}_3, \boldsymbol{y}_3)) = \Delta(\mathcal{E}_1, \mathcal{E}_3) + \delta(\boldsymbol{y}_1, \boldsymbol{y}_3) \tag{23}$$

$$\leq \Delta(\mathcal{E}_1, \mathcal{E}_2) + \Delta(\mathcal{E}_2, \mathcal{E}_3) + \delta(\boldsymbol{y}_1, \boldsymbol{y}_2) + \delta(\boldsymbol{y}_2, \boldsymbol{y}_3) \tag{24}$$

$$= d((\mathcal{E}_1, \boldsymbol{y}_1), (\mathcal{E}_2, \boldsymbol{y}_2)) + d((\mathcal{E}_2, \boldsymbol{y}_2), (\mathcal{E}_3, \boldsymbol{y}_3)) \tag{25}$$

So $d((\mathcal{E}, \boldsymbol{y}), (\tilde{\mathcal{E}}, \tilde{\boldsymbol{y}})$ is indeed a distance metric.

$\square$

---

**Algorithm 3** Efficient Edge Candidate Selection for Link Augmentation

---

**Input:** Number of nodes per class $n_c$, node confidence scores per class:
  $\{S_c = [s_{c,1}, s_{c,2}, \ldots, s_{c,n_c}]\}$ sorted descending, desired number of edges to add $\varepsilon \cdot e$
**Output:** Top-$\varepsilon \cdot e$ pseudo-edges with highest sum of confidence scores
 1: Initialize global max-heap $G$ (size = number of classes)
 2: **for** each class $c$ **do**
 3:    Initialize empty local max-heap $L_c$
 4:    Insert initial pair $(0, 1)$ with score $s = S_c[u] + S_c[v]$ and class index as tuple $(u, v, s, c)$ into $G$
 5: **end for**
 6: **while** number of added edges smaller than $\varepsilon \cdot e$ **do**
 7:    Extract top $(u, v, s, c)$ from global heap $G$
 8:    Add $e = (u, v)$ to candidate edge set
 9:    For local heap $L_c$, consider children pairs:
10:      $child_1 = (u, v + 1)$ if $v + 1 \leq n_c$
11:      $child_2 = (u + 1, v)$ if $u + 2 = v$
12:    **for** each valid child pair $(u, v)$ **do**
13:      Compute sum of score $s = S_c[u] + S_c[v]$
14:      Insert tuple $(u, v, s)$ into $L_c$
15:    **end for**
16:    Extract next top tuple $(u', v', s)$ from $L_c$
17:    Insert $(u', v', s, c)$ into global heap $G$
18: **end while**
19: **return** Final candidate edges across all classes

---

# D. Algorithms

## D.1. Efficient Selection of Pseudo-Edge Candidates

As the number of nodes, $k$ or $\varepsilon$ goes up, the number of possible edge pairs also scales up, which results in significant computational overhead. To address this challenge, we propose a time and memory-efficient algorithm for selecting edge candidates with the highest combined confidence scores across all classes. The algorithm for selecting edges is described in Algo. 3. A global max-heap keeps track of which class has the next best edge candidate according to the sum of scores. We initialize the global heap by inserting the first pair $(0, 1)$ from each class, as these pairs hold the maximum possible sum of scores within their respective classes. To add edges, we repeatedly pick the top element from the global heap until the number of edges selected exceeds the threshold defined by $\varepsilon$.

To supply the global heap with the best candidates from each class, every class maintains a local heap, which is initially empty. Each time an edge is selected from the global heap, the algorithm accesses the corresponding class's local heap and inserts the next candidate pairs. To prevent local heaps from becoming prohibitively large, we do not insert all possible node pairs at once. Instead, node pairs are added only when they can potentially represent the best candidate. This is enabled by pre-sorting nodes and their confidence scores within each class in descending order. Consequently, nodes with smaller indices correspond to higher scores. This ordering induces a binary tree structure over node pairs, illustrated in Fig. 16, where an edge $(u, v)$ has a larger combined score than $(u, v+1)$, and $(u+1, v)$. As the criterion, the sum of scores, is commutative, $(u, v)$ and $(v, u)$ represent the same edge candidate, so we consider only node pairs $(u, v)$ where $u < v$. This results in the half binary tree structure in Fig. 16.

Layer

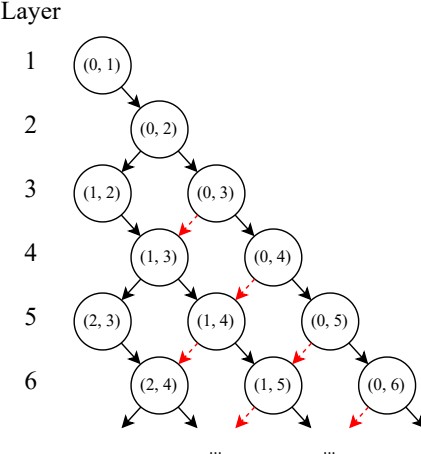

*Figure 16.* Tree structure for spawning edge candidates. An edge in layer $i$ is guaranteed to have a higher score than in layer $i + 1$ and a lower score than in layer $i - 1$.

Since most nodes have two parents, simply adding both $(u, v + 1)$ and $(u + 1, v)$ to the local heap would result in duplicate insertions of many nodes. To avoid this, we add the pair $(u + 1, v)$ only when $u + 2 = v$. This condition effectively removes the red dashed connections shown in Fig. 16 and ensures that each edge pair is added to the local heap exactly once. This also guarantees that the local heap always contains the node pair with the highest possible sum of scores.

### D.2. Proposed General Certification Pipeline Against Both Label Flipping and Structure Perturbations

Algo. 4 shows the joint certification pipeline for ST-GPA. Regarding the training order, we do edge generation first, since we find effective co-training relies on meaningful graph structure. Further, we assume the number of label partitions $k_y$ and structure partitions $k_e$ are the same, i.e., $k = k_e = k_y$. Otherwise, the partitioning in Lines 1-2 has to be adapted to duplicate the objects associated to the smaller partition number into all partitions, affecting $d_h$ when calculating the certificate using the base classifier counts based on Thm. 3.1.

---

**Algorithm 4** ST-GPA joint pipeline

---

**Input:** Graph dataset $D = (\mathcal{E}, \mathcal{X}, \mathcal{Y})$, a label propagation method $\hat{y} = LP(e_i, \mathcal{X}, y)$, a label self-training method $\hat{y} = LST(e_i, \mathcal{X}, y)$, an link prediction method $\hat{e}_i = EST(e_i, \mathcal{X}, y)$, a training order $o_{trn} = (op_1, \ldots, op_R) \in \{LP, LST, EST\}^R$, number of partitions $k$

**Output:** An ensemble classifier prediction and the associated certificate for a test node $v$.

1: Split labels into partitions $\boldsymbol{y}_i = \{y | y \in \mathcal{Y}, h(\mathcal{X}, y) \equiv i \pmod{k}\}$
2: Split edges into partitions $e_i = \{e | e \in \mathcal{E}, h(\mathcal{X}, e) \equiv i \pmod{k}\}$
3: **for** each operation $op$ in the given order $o_{trn}$ **do**
4:     **for** each partition $i$ **do**
5:         $\hat{e}_i$ or $\hat{y} = op(e_i, \mathcal{X}, \boldsymbol{y}_i)$
6:         $e_i = e_i \cup \hat{e}_i$ or $\boldsymbol{y}_i = \boldsymbol{y}_i \cup \hat{y}$
7:     **end for**
8: **end for**
9: Train base classifiers $f_i$ for each partition $i$ on final edges, labels and features, i.e., on $\hat{D} = \{e_i, \mathcal{X}, \boldsymbol{y}_i\}$.
10: Count base classifier predictions $n_c(v)$ for all $c \in [C]$
11: Obtain majority class $c^* = \arg\max_{c \in [C]} n_c(v)$
12: Calculate robust margin (see Thm. 3.1)
    $r_m = \lfloor (n_{c^*} - \max_{c' \neq c^*} (n_{c'}(v) + \mathbb{1}_{c' < c^*}))/2 \rfloor$
13: **return** $c^*, r_m$

---

# E. Comparison to Existing Works

## E.1. Comparison to LabelCert for Label Flipping

LabelCert (Sabanayagam et al., 2025) introduce the first exact certification method for GNNs against label poisoning, leveraging the Neural Tangent Kernel to reformulate the bilevel poisoning problem as a Mixed-Integer Linear Program. While their certificates are exact in the infinite-width limit, the MILP-based approach is computationally tractable only for binary classification tasks with a very small number of labels (20-40 in total), which represents a highly constrained experimental regime. Evaluation on a binary class Cora-ML dataset (Table 4) reveals that our method achieves certified accuracies that are remarkably competitive despite this sub-optimal setting for label partitioning. Specifically, our approach matches LabelCert within a small margin at a 5% perturbation budget and outperforms it at 10% perturbations. Furthermore, our method scales to graph datasets several orders of magnitude larger in both number of nodes and number of label classes than what LabelCert can handle, demonstrating a practical advantage that extends well beyond the narrow binary setting in which exact MILP-based certification is feasible.

*Table 4.* Certified accuracy (%) on Cora-ML (binary) under label poisoning, comparing our method to LabelCert.

| Method | Label Perturbations | |
|---|---|---|
| | **5%** | **10%** |
| Ours (co-train then self-train) | $66.5 \pm 17.8$ | $36.2 \pm 1.4$ |
| LabelCert | $68.7 \pm 6.8$ | $32.6 \pm 2.0$ |

## E.2. Discussion on the Relation to PGNNCert Against Structure Perturbations

Li et al. (2025) apply a similar partitioning scheme PGNNCert to derive poisoning certificates for GNNs. In their work, a threat model of arbitrarily perturbing edges, nodes, and node features (but not labels) is considered. In this section, we provide comparison between our results and theirs. Due to that their certified accuracy does not outperform an infinitely robust MLP, we do not report them in Sec. 6.

To compare our works directly, we adopted their published code and use their exact dataset splits. We then run our proposed approach on their data split. Note that the only difference in the setup is the model and hyperparameters for training, where we use a 2-layered GCN as described in App. A, and they use a 3-layered GCN with skip connections and linear layer. In PGNNCert, 2 different partitioning schemes were proposed, namely edge centric and node centric, which has similar performance against structure perturbations, so we report the result with the node centric variant only. Due to the limitation of implementation of PGNNCert, their code does not scale to larger number of partitions due to memory limits, so we choose $k = 60$ as a compromise and keep other parameters exactly the same for a fair comparison. Note that normally our link prediction scheme can be used with significantly larger $k$-s to generate competitive certificates, so we also report $k = 600$ with the same setup. Although PGNNCert doesn't report results on Cora, their code can be easily adapted as the Planetoid datasets share the same data loader in PyTorch, so we report results on Cora as well.

As shown in Fig. 17, PGNNCert-N's performance is consistently outperformed by the infinitely robust MLP baseline on all datasets. In the mean time, our link prediction method shows similar improvements to certified accuracy over the MLP baseline as previously reported in Sec. 6 and App. B. At $k = 60$, the allowed perturbation budget is very small. But the reported $k = 600$ curves shows the normal performance of our method with more partitions, yielding larger perturbation budgets.

Provided that PGNNCert's performance is below the MLP baseline in the case of structure perturbation, we don't report a comparison to PGNNCert in the results in this paper, as we already use the MLP baseline as a lowest acceptable case in all our results against structure perturbation.

In PGNNCert, no certificate against label flipping is reported. As we have already shown in App. B.1, the partitioning scheme does not readily transfer to labels, so we don't compare our results with PGNNCert on label flipping.

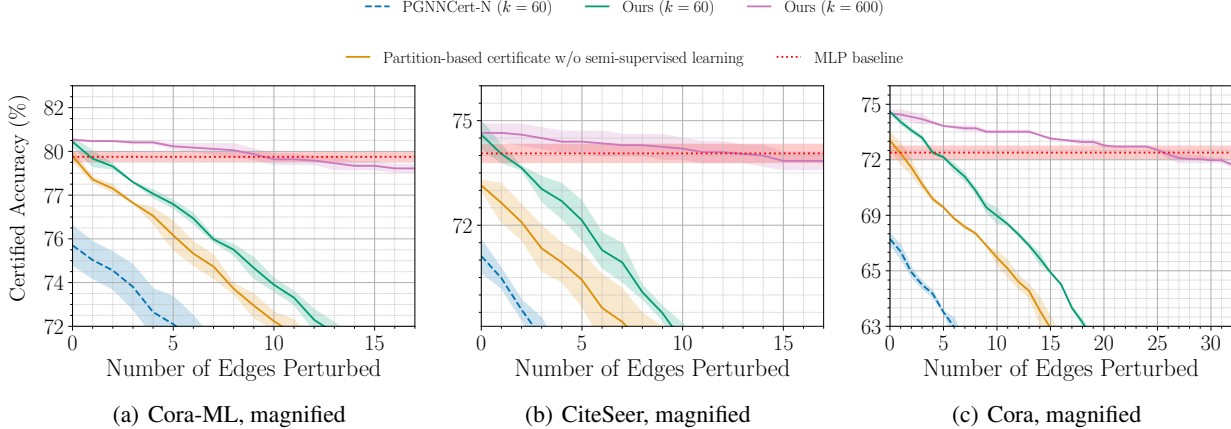

*Figure 17.* Certified accuracies of our method compared to Li et al. (2025), $\varepsilon = 30$. PGNNCert-N's performance falls steadily below the MLP baseline which is infinitely robust to structure perturbation; a trivially applied partitioning scheme has only on-par clean accuracy with an MLP, yielding hardly any meaningful perturbation budget; with our proposed link prediction method, the certified accuracies is most competitive.

## F. Scalability of Our Methods

In this section, we analyze the scalability of our three proposed self-training methods by reporting the relative time used in our experiments. In all our experiments, we use a single NVIDIA GTX1080Ti GPU. The link prediction uses CPU only. While it can be easily parallelized, we report the performance on 1 CPU core only, as link prediction on individual partitions are usually done so. The times reported are for relative reference, considering the number of nodes and edges in each dataset.

We first point out that the time complexity scales linearly with the number of partitions $k$, since the partitions are disjoint once they are created. Partitioning the dataset takes a negligible amount of time, and can easily be accelerated by pre-processing the dataset and storing all the partition indices. Consequently, it is reasonable that we only report the time taken per partition in this section. The memory complexity is irrelevant w.r.t. $k$ by the same reason that the partitions are disjoint. The training on individual partitions could be done in a parallel or distributed manner.

*Table 5.* Time used for co-training and self-training with $t = 50$

|  | Cora-ML | CiteSeer | Cora | PubMed | Wiki-CS | ogbn-arxiv |
|---|---|---|---|---|---|---|
| graph size (Nr. nodes) | 2810 | 2110 | 2708 | 19717 | 11701 | 169343 |
| **training time per partition (s)** | **6.4** | **5.5** | **5.5** | **2.7** | **3.1** | **40.9** |
| thereof training w/o SSL | 36.1% | 37.3% | 36.4% | 31.9% | 29.0% | 31.4% |
| thereof C/T | 26.8% | 24.7% | 25.2% | 30.8% | 36.4% | 37.0% |
| thereof S/T | 37.1% | 38.0% | 38.4% | 37.3% | 34.6% | 31.6% |

Table 5 shows the time taken to train an individual partition on all datasets we tested. Label partition training is generally very fast. Note that the time portion reported for training without semi-supervised learning represents the time baseline for training on the raw partition, which has also similar time complexity of training on the clean graph. Judging from the percentages reported, co-training and self-training is just another round of training of a GNN, which takes similar time as training the first one on the raw partition. The results show good scalability of our label self-training methods as the time portion stays roughly equal regardless of the size of the dataset, and the total time necessary to train the ensemble scales roughly linearly with the graph size (number of nodes).

Table 6 shows the time taken per partition for edge semi-supervised training. Although smaller datasets has faster training time per partition, more partitions are required in structure perturbation to generate a meaningful certificate. Therefore, the time taken to train an edge ensemble is usually comparably longer than a label ensemble. In the smaller datasets, training

*Table 6.* Time used for link prediction with $\varepsilon = 1.0$

|  | Cora-ML | CiteSeer | Cora | PubMed | Wiki-CS | ogbn-arxiv |
| --- | --- | --- | --- | --- | --- | --- |
| graph size (Nr. edges) | 7981 | 3668 | 5069 | 44324 | 216343 | 1166243 |
| **training time per partition (s)** | **1.9** | **1.0** | **2.0** | **9.3** | **19.3** | **67.8** |
| thereof training w/o SSL | 32.3% | 58.7% | 57.5% | 14.4% | 8.2% | 6.6% |
| thereof L/P | 7.3% | 3.8% | 5.5% | 13.8% | 34.5% | 42.8% |
| thereof training with pseudo-edges | 60.4% | 37.5% | 37.0% | 71.8% | 57.2% | 50.6% |

without and with the pseudo-edges take roughly the same time. However, as the number of edges grows, training with pseudo-edges becomes the predominant factor in total training time, and the time taken to find the pseudo-edges becomes non-negligible. On larger graphs, this is especially a choking factor as we usually add more pseudo-edges than there originally are ($\varepsilon \gg 1.0$), making the link prediction very time-consuming.

## G. Empirical Tightness and Transfer Attacks

Since our method provides exact certification that already incorporates the lower bound on robustness, a key open question is how tight these certificates are in practice. To investigate this empirically, we conduct a transfer attack using Meta-PGD from Mujkanovic et al. (2022), transferring adversarial edges crafted against a GCN surrogate to our partition-and-ensemble framework. We note that gradient-based attacks such as PR-BCD (Mujkanovic et al., 2022) cannot be directly applied to ensemble-based methods, since doing so would require differentiating through the random partitioning and unrolling the training of many base classifiers on distinct graph subsets, a significant technical challenge for which no established solution currently exists. Shown in Fig. 18, the transfer attack proves largely ineffective: robust accuracy remains stable at $k = 800$. We attribute this to two factors: the non-adaptiveness of the attack, which is optimized against a single GCN rather than the ensemble, and the strong graph partitioning inherent to our method, which dilutes the influence of any fixed set of adversarial edges across independently trained base classifiers. Here we want to note the findings from Mohgaonkar et al. (2026), who demonstrate on image datasets that DPA-based methods for label certification tend to be overly conservative for small $k$, but approach tightness for larger $k$. Thus, one could expect similar behavior for graphs.

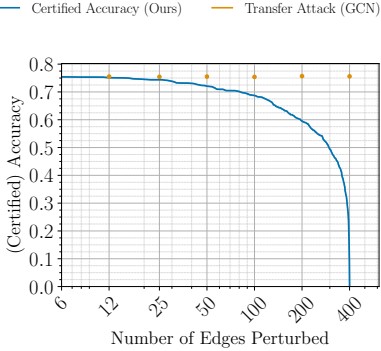

*Figure 18.* Performing a transfer attack from a GCN using Meta-PGD for structural poisoning. We find the transfer attack from a GCN to be mostly ineffective in reducing the accuracy of the ensemble ($k = 800$ for both, certified and robust accuracy).

These results motivate an important open problem: the development of *adaptive* poisoning attacks tailored specifically to partition-and-ensemble methods. Such attacks would need to simultaneously optimize adversarial edges with respect to the partitioning scheme and the ensemble decision rule, which is a non-trivial objective given the discrete and stochastic nature of graph partitioning. Progress in this direction would enable a rigorous empirical assessment of certificate tightness beyond what transfer attacks can currently provide, and we view it as an important avenue for future work.

## H. Discussion on the Impact of Homophily

In this section, we conduct a systematic study on Contextual Stochastic Block Models (CSBMs), investigating what threshold of homophily our certification method requires for good empirical results. We vary homophily from 0% (pure heterophily) to 100% (pure homophily) in terms of the percentage of edges that connect same-class nodes. We find that both our certificates against label and structure perturbations start to provide benefits from 60% homophily, with strong results when the homophily exceeds 70%. (For context, Cora has 81% homophily.) The detailed results are shown in Figs. 19 and 20, and we refer to the figure captions for details on the experimental setup.

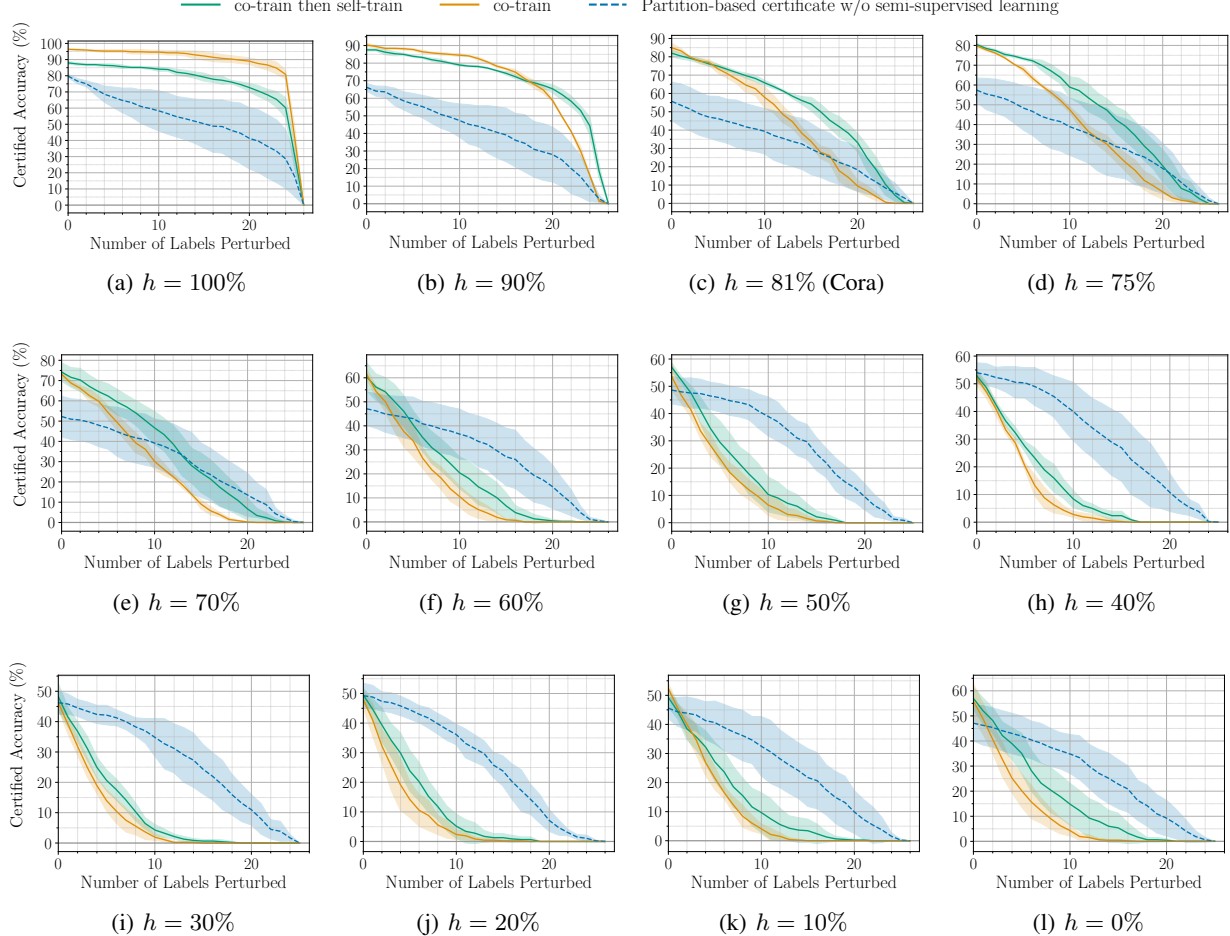

*Figure 19.* Label flipping certificate results with $k = 50$ on CSBM graphs with different levels of homophily. $h$ = percentage of edges that connect same class nodes. CSBM: 774 nodes (same number of nodes/class and edge density as Cora). Other parametrization following Gosch et al. (2023) with $K = 1.5$, and $\sigma = 1$. We use 60% nodes for training and 20% for validation and testing respectively.

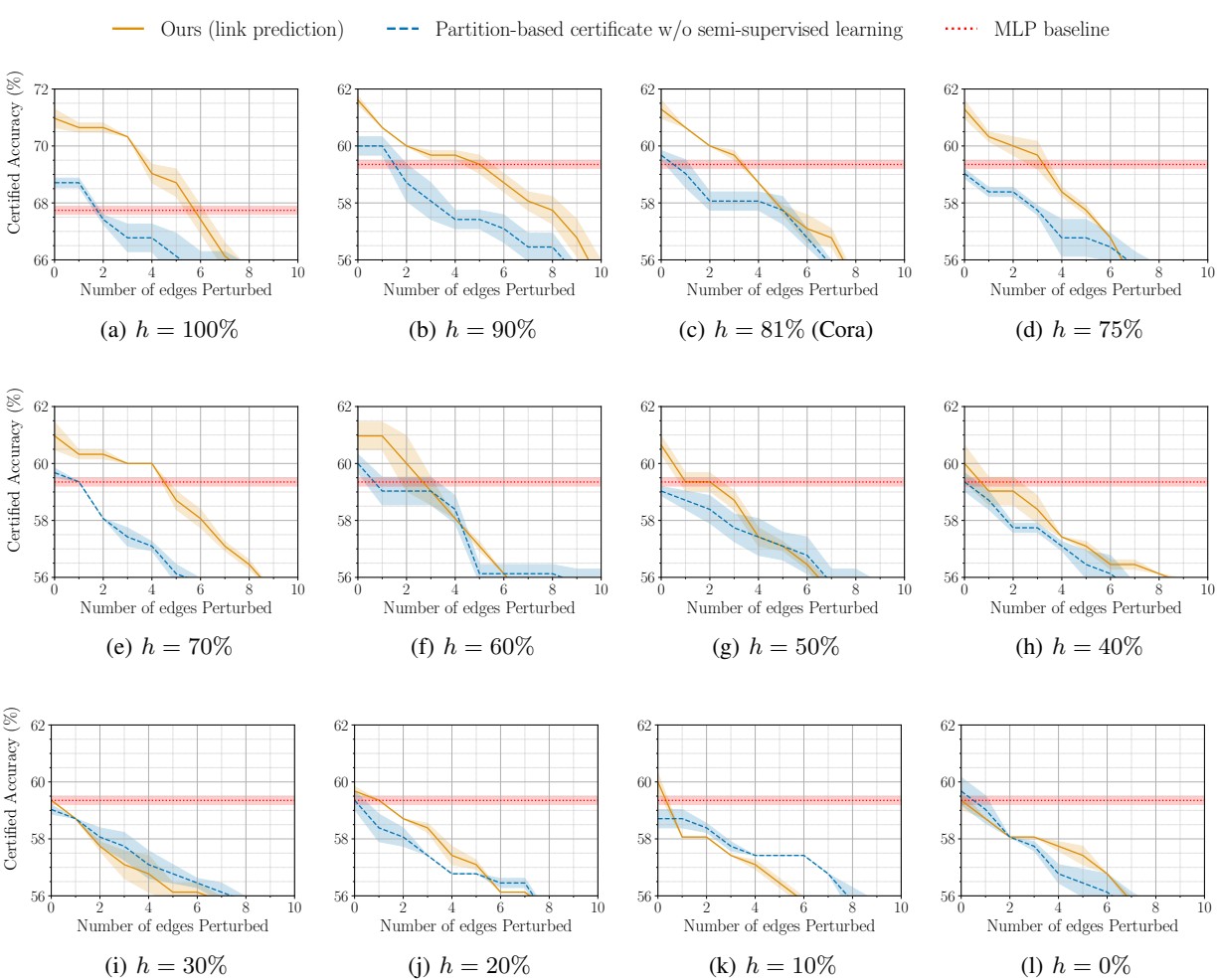

*Figure 20.* Our structural certificate on CSBM graphs with diff level of homophily. $h$ = percentage of edges that connect same class nodes. CSBM: 774 nodes (same number of nodes per class as Cora), other parametrization following Gosch et al. (2023) with $K = 1.0$, and $\sigma = 1$. For certification we use $k = 100$ partitions and $\varepsilon = 1$. 60% nodes are used for training and 20% for validation and testing respectively.

