# OpenReview forum: "Certifying Graph Neural Networks Against Label and Structure Poisoning"
_ICML.cc/2026/Conference — ICML 2026 regular_

### Official Review · Reviewer_7Vyy · 2026-03-11

**Soundness:** 2
**Presentation:** 2
**Significance:** 2
**Originality:** 2
**Overall Recommendation:** 4
**Confidence:** 3

**Summary:**

The paper introduces deep Self-Training Graph Partition Aggregation (ST-GPA), a novel semi-supervised learning framework designed to provide certified robustness for Graph Neural Networks (GNNs) against label and structure poisoning attacks. The authors identify that existing partition-and-aggregate schemes, which are effective in the image domain, fail when applied to graphs due to inherent label and structure sparsity. To address this, the ST-GPA framework enriches the partitioned, sparse subgraphs by generating synthetic labels and edges utilizing self-training, label propagation, and link prediction. Extensive evaluations across multiple datasets demonstrate that this architecture-agnostic approach significantly improves certified accuracy against poisoning attacks while maintaining strong clean accuracy.

**Compliance With Llm Reviewing Policy:**

Affirmed.

**Final Justification:**

Thank you for addressing my concerns, and I will retain my original score.

**Key Questions For Authors:**

Questions:
1. The hyperparameter ϵ dictates the addition of pseudo-edges within partitions. How sensitive is the certified robustness to the variance of this parameter across datasets with vastly different underlying structural densities?
2. Given the scalability issues associated with the current link prediction method, could more lightweight, sparsity-aware link prediction algorithms be integrated to reduce the time complexity for large-scale graphs?

**Limitations:**

yes

**Strengths And Weaknesses:**

Strengths:
1. The manuscript clearly identifies and characterizes a critical failure, structural and label sparsity, that occurs when directly applying traditional partitioning certification methods to graph data.
2. The proposed ST-GPA framework is an innovative application of semi-supervised learning. By explicitly generating pseudo-labels and synthetic edges, it directly mitigates the fundamental sparsity constraints within graph partitions.
3. The experimental evaluation is robust, demonstrating state-of-the-art certified poisoning robustness across multiple benchmark datasets and different GNN architectures.
Weaknesses:
1. The link prediction component introduces considerable computational and memory overhead, making the certification process difficult to scale efficiently to very large or dense datasets like ogbn-arxiv.
2. The framework relies fundamentally on the homophily assumption, which restricts its applicability and effectiveness on heterophilic graphs.

---

> ### Author Rebuttal · Authors · 2026-03-31
>
> We want to thank the reviewer for the constructive feedback! Below, we address all the raised points in detail.
>
> > W1. The link prediction component introduces considerable computational and memory overhead, making the certification process difficult to scale efficiently to very large or dense datasets like ogbn-arxiv.
>
> We recognize that the link prediction component for structure certificates faces increased computational demands on very large or dense graphs, as we also openly discuss in the limitation section of our paper. However, please note that our method is particularly tailored to solve this challenging problem for sparse graphs, where it performs strongly. We believe this constitutes an important contribution, in particular since prior approaches were not able to provide non-trivial guarantees for any setting. In response to your comment, we will further discuss these limitations in more detail in the final version of the paper. Note that our label certificate scales well to large and dense graphs such as ogbn-arxiv (see Figure 12, Page 15 and Figure 2 here: https://figshare.com/s/ce37834b03144a1bf940), as well as to the even more dense graphs Amazon Photo & Amazon Computers (see Figure 1 in the linked document).
>
>
> > W2. The framework relies fundamentally on the homophily assumption, which restricts its applicability and effectiveness on heterophilic graphs.
>
> We agree that our current co- and self-training, and link-prediction components depend on homophily, as we acknowledge and discuss in the limitations section of our paper. However, we now include a detailed study on the homophily assumption on synthetic graphs, where we vary homophily from 0% (pure heterophily) to 100% (pure homophily) to quantify this dependence. We find that both our certificates against label and structure perturbations start to provide benefits from 60% homophily, with strong results for >70% homophily. Details can be found in https://figshare.com/s/ce37834b03144a1bf940 in Figures 4 and 5. As prior work did not achieve any non-trivial guarantees for structure or label poisoning even for homophilic graphs, we think it is an important contribution to have the first practically effective certificate for graphs with homophily $\ge60$%, and we are convinced that we thereby lay a solid foundation for follow-up works expanding the applicability of graph-based poisoning certificates.
>
> > Q1. The hyperparameter ϵ dictates the addition of pseudo-edges within partitions. How sensitive is the certified robustness to the variance of this parameter across datasets with vastly different underlying structural densities?
>
> We do find that $\epsilon$ is a highly dataset-specific parameter. We investigate the sensitivity of our structure certificate to $\epsilon$ in Figure 14 (Page 15) in our paper and find that even for structurally more similar datasets, the optimal $\epsilon$ can vary significantly. Thus, one can conclude that if the structural densities differ significantly, the optimal $\epsilon$ will too.
>
> > Q2. Given the scalability issues associated with the current link prediction method, could more lightweight, sparsity-aware link prediction algorithms be integrated to reduce the time complexity for large-scale graphs?
>
> Yes. Indeed, we think this could be one way forward to reduce the computational complexity of the link prediction task. In the preparation of our method, we tried several link prediction methods from the literature. However, we found limited success, which we attribute to the extreme sparsity in each graph induced through the strong partitioning. Thus, we would be interested in suggestions on further concrete and interesting works on this topic.

---

> > ### Author Rebuttal · Reviewer_7Vyy · 2026-04-02
> >
> > You mentioned that extremely large heterogeneous graphs such as ogbn-mag are beyond the scope of this work. Could you please clarify the key technical obstacles that prevent your method from being extended to such graphs? Meanwhile, you stated that existing approaches cannot provide meaningful robustness guarantees in this scenario. We wonder whether your own method also faces the same issue. Could you theoretically analyze the computational and memory overhead of your proposed approach, so that readers can better understand its applicable boundaries and scalability challenges?
> > We are also concerned that the computational cost may be prohibitively high to conduct a complete evaluation, which could affect the significance of extending the method to such settings.

---

> > > ### Author Response · Authors · 2026-04-04
> > >
> > > We want to thank the reviewer for the continued discussion! In the following, we carefully address all remaining concerns.
> > >
> > > > You mentioned that extremely large heterogeneous graphs such as ogbn-mag are beyond the scope of this work. Could you please clarify the key technical obstacles that prevent your method from being extended to such graphs?
> > >
> > > The key technical obstacles are two-fold: The underlying prediction model (for which we use GCN and similar GNNs) needs to be made aware and tuned to the heterogeneity of the graph. For `ogbn-mag`, this would mean being aware of 4 different node types and 4 different edge types. Then, due to the size of the graph, the training code has to be adapted to batch-wise training and evaluation. Therefore, we would like to clarify that very large graphs such as `ogbn-mag` (with millions of nodes) are not, in principle, outside the boundaries of our method. However, extending the empirical evaluation to such extremely large datasets requires significant adaptations of our model, evaluation, and training code, which, together with all the experiments we provided during this rebuttal, we see as beyond the scope of the rebuttal period, and not as the focus of this work.
> > >
> > > > Meanwhile, you stated that existing approaches cannot provide meaningful robustness guarantees in this scenario. We wonder whether your own method also faces the same issue.
> > >
> > > No, our own method does provide meaningful robustness guarantees as shown in Figures 4, 5 & 6 in the Results Section of our paper. In particular, existing methods fail to provide meaningful robustness guarantees not just on large but even on standard benchmark graphs. The primary reason is not scalability, but their inability to account for sparsity (a central challenge in graph domains). In contrast, our method explicitly accounts for sparsity and yields non-trivial guarantees against both structure and label poisoning across many different graph datasets (see the above-mentioned figures and new results in https://figshare.com/s/ce37834b03144a1bf940). As this was not achieved before, we believe our contributions are significant and provide valuable insights for the graph learning and robustness communities.
> > >
> > > > Could you theoretically analyze the computational and memory overhead of your proposed approach, so that readers can better understand its applicable boundaries and scalability challenges? We are also concerned that the computational cost may be prohibitively high to conduct a complete evaluation, which could affect the significance of extending the method to such settings.
> > >
> > > We want to point the reviewer to our detailed theoretical and empirical discussion of the computational and memory overhead found in Appendix F (Page 20), where we also discuss applications to larger graphs, including time measurements for the large `ogbn-arxiv` dataset. In particular, since training on disjoint partitions can be done in parallel, our overall approach is primarily constrained by the time and memory required to train a single model on one partition; and the time for one partition scales roughly linearly with the number of nodes (for details, see Appendix F). Thus, we want to emphasize that the computational cost does not inherently limit our method's applicability to even larger graphs, though we do not see this as the focus of our work. As we explained in the answer to Reviewer 7PuY, our empirical analysis already covers a wide range of graphs with different structures, including denser and larger graphs such as `ogbn-arxiv` (next to Appendix F, see also Figures 1.1 and 1.2 in https://figshare.com/s/ce37834b03144a1bf940). As prior approaches were not able to provide non-trivial structure or label poisoning guarantees for any of these datasets and settings, we think this is an important contribution. In the revised manuscript, we now plan to include the discussion on applying our method to heterogeneous graphs, next to large graphs already discussed in Appendix F.
> > >
> > > We are confident to have addressed the remaining concerns of the reviewer and are happy about the continued discussion!
> > >
> > > **EDIT 7.4.:** Improved Clarity of Answers.

---

### Official Review · Reviewer_7PuY · 2026-03-12

**Soundness:** 3
**Presentation:** 3
**Significance:** 2
**Originality:** 3
**Overall Recommendation:** 4
**Confidence:** 4

**Summary:**

The paper shows that directly utilizing popular robustness method from the image domain and applying to graph learning tasks fail owing to label/structure sparsity after data partitioning. To address this, the paper proposes ST-GPA, a semi-supervised learning framework that augments partitions with pseudo-labels and/or synthetic edges and enable certified poisoning robustness for GNNs.

**Compliance With Llm Reviewing Policy:**

Affirmed.

**Final Justification:**

The additional experiments presented by the authors during rebuttal provides additional insights about the proposed model; the SBM experiments highlight both the strengths and limitations of the proposed of ST-GPA. Overall, I incline towards accepting this paper.

**Key Questions For Authors:**

1. Did the authors try applying their framework to node classification tasks with higher class labels than 40? This could either include a synthetic graph or existing datasets like ogbn-mag.
2. See weaknesses above.

**Limitations:**

yes

**Strengths And Weaknesses:**

**Strengths**
1. The paper is well written. It exposes the limitations of directly applying poisoning robustness guarantees based on data partitioning and aggregation to graph structures.
2. To addresses the previous shortcoming, the paper generalizes the partition and aggregate robustness guarantees to non-i.i.d structured data.
3. The proposed framework, ST-GPA, addresses the sparsity issue in graph partitioning by augmenting pseudo labels/synthetic edges. Empirical experiments over homophilic graphs validate the effectiveness of the proposed framework.

**Weaknesses**
1. While noted in limitations that the proposed framework primarily works for homophilic graphs, it is unclear what threshold of homophily is required for the assumptions to hold. It would help if the authors conducted experiments on synthetic graphs with increasing levels of homophily, total nodes/edges/classes to empirically establish the generality of the robustness.
2. The choice of GNNs used in the experiments still feature old and popular methods. It would help strengthen the claims if the authors also included more recent GNN baselines like [1] (and including a transformer based GNN e.g., [2])

---

**References**

[1] Revisiting Heterophily For Graph Neural Networks, Luan et. al., NeurIPS 2022

[2] Rethinking Tokenized Graph Transformers for Node Classification, Chen et. al., NeurIPS 2025

---

> ### Author Rebuttal · Authors · 2026-03-31
>
> We want to thank the reviewer for the constructive feedback! We have now systematically studied the homophily dependence and added a modern graph transformer architecture. In particular, we address all mentioned points in detail below.
>
> > W1 i). While noted in limitations that the proposed framework primarily works for homophilic graphs, it is unclear what threshold of homophily is required for the assumptions to hold. It would help if the authors conducted experiments on synthetic graphs with increasing levels of homophily [...] to empirically establish the generality of the robustness.
>
> We now **include a systematic study** on Contextual Stochastic Block Models (CSBMs), investigating what **threshold of homophily** our method requires for good empirical results. We vary homophily from 0% (pure heterophily) to 100% (pure homophily). We find that both our certificates against label and structure perturbations start to provide benefits from 60% homophily, with strong results for >70% homophily. (Exemplary, Cora has 81% homophily.) The detailed results can be found in https://figshare.com/s/ce37834b03144a1bf940  in Figures 4 and 5.
>
> > W1 ii). [...] experiments [...] total nodes/edges/classes & Q1. Did the authors try applying their framework to node classification tasks with higher class labels than 40? This could either include a synthetic graph or existing datasets like ogbn-mag.
>
> We did not include experiments beyond 40 classes or on ogbn-mag. The primary reason is not about the number of classes, but the joint edge-heterogeneity and very large-scale nature of graphs such as ogbn-mag, which lies outside the intended scope of this work. Including the new results, we investigate graph datasets from **778 to 169,343 nodes**, from **3,014 to 2,315,598 edges**, and from **2 to 40 classes**. Together with the new CSBM study, we think this well marks the relevant regime of homophilic graphs where our certificate works well. As prior approaches were not able to provide non-trivial structure or label poisoning guarantees for any of these datasets and settings, we think this is an important contribution. This is the reason why we see an extension of our certificate to heterogeneous and extremely large graphs with millions of nodes such as `ogbn-mag` as out of scope for this work. However, we are convinced that we laid a solid foundation for making progress in this field, with the ultimate goal to scale and be applicable to the most diverse graph settings.
>
> > W2. The choice of GNNs used in the experiments still feature old and popular methods. It would help strengthen the claims if the authors also included more recent GNN baselines like [1] (and including a transformer based GNN e.g., [2]).
> > [1] Revisiting Heterophily For Graph Neural Networks, Luan et. al., NeurIPS 2022.
> > [2] Rethinking Tokenized Graph Transformers for Node Classification, Chen et. al., NeurIPS 2025.
>
> We now **include results for a Graph Transformer (GT)**, in particular NodeFormer [1], because it is a well-established transformer baseline with public code available and thus, easier to evaluate within the rebuttal timeline. The results (see Figure 3 in https://figshare.com/s/ce37834b03144a1bf940) highlight that our certification strategy works well with a modern GT architecture. In particular, we find that NodeFormer w/o our semi-supervised learning strategies struggles to learn from sparsely labeled partitions. However, enriching the partitions with pseudo-labels using our proposed co-training scheme leads to good clean accuracies and certified robustness of the GT-ensemble against label poisoning.
>
> Note that a GCN is still a strong and competitive baseline for node classification, often outperforming more modern architectures including GTs given the correct tuning of the model [2]. In particular, for larger datasets (ogbn-arxiv, and the newly added Amazon-Photo and Amazon-Computers), we follow the hyperparameter tuning scheme from [2].
>
> [1] Wu et al. "NodeFormer: A Scalable Graph Structure Learning Transformer for Node Classification", NeurIPS 2022
> [2] Luo et al. "Classic GNNs are Strong Baselines: Reassessing GNNs for Node Classification", NeurIPS 2024

---

> > ### Author Rebuttal · Reviewer_7PuY · 2026-04-02
> >
> > Thank you for the detailed and thoughtful rebuttal; I appreciate the clarifications provided.
> > Per ICML guidelines, rebuttals should be self-contained and reviewers are not expected to follow external URLs, so I do not consider evidence provided via external links.
> >
> > I will maintain my current rating. The additional experiments improve the clarity of the paper. However, I do not assign a higher score as the synthetic experiments indicate strong performance primarily for homophily scores above 70%, which may limit practical applicability. Nonetheless, I retain a positive assessment overall.

---

> > > ### Author Response · Authors · 2026-04-08
> > >
> > > We want to thank the reviewer again for their constructive review and for acknowledging that their concerns have been adequately addressed! We will include all additional experiments and results in the paper. As non-trivial structure or label poisoning certificates for graphs have not been achieved prior to this work, not even for homophilic settings common in practice, we think this constitutes an important contribution with practical applicability. Systemically investigating the boundaries of settings where our method works, as asked by the reviewer and provided in our rebuttal, further improves on the practical applicability by making explicit the significant range of graph setting where our methods performs well, and were future work can draw on and begin to further push the boundaries of what is currently possible in poisoning certification for graphs.

---

### Official Review · Reviewer_hbbX · 2026-03-12

**Soundness:** 3
**Presentation:** 3
**Significance:** 3
**Originality:** 2
**Overall Recommendation:** 4
**Confidence:** 3

**Summary:**

The paper focuses on data poisoning attacks for Graph Neural Networks (GNNs). In particular, they propose a new, certified defense against label and structure poisoning for node-level GNNs. They introduce a framework called Self-Training Graph Partition Aggregation and evaluate it experimentally.

**Compliance With Llm Reviewing Policy:**

Affirmed.

**Final Justification:**

The rebuttal answers addressed my main concerns. I maintain my positive score.

**Key Questions For Authors:**

- The paper mentions that "our insights ...  may be of interest beyond graph learning". Can you provide some concrete examples for applications (or potential experiments to run) for different domains?
- Can you provide a comparison with other works' baselines and elaborate further on why they are not adaptable? Even running and seeing a very poor result compared to this work would be valuable if verified experimentally.

**Limitations:**

yes

**Strengths And Weaknesses:**

Strengths:
- The paper covers an important topic - data poisoning attacks for GNNs.'
- They base their solution on Deep Partition Aggregation (DPA; Levine & Fezi, 2021). However, they correctly identify its limitations for the graph domain, and propose their extended solution - Generalized DPA.
- The Appendix includes a proof for the Generalized DPA Theorem.
- The proposed method is interesting, and each component is tested with ablations.
- The experiments show decent results in every studied setting.

Weaknesses:
- The experiments were run on only 4 fairly small datasets.
- Section Related Works mentions multiple graph-domain certifiable defenses. The authors claim that none of them is applicable to this setting (each of the works is either not applicable to the perturbation models, or infinite-width GNNs, or scaling only to very small datasets). As a result, the authors only compare their method to an MLP baseline, which seems very simple. I would double-check if any of the related works can be extended or adapted to compare the results.

---

> ### Author Rebuttal · Authors · 2026-03-31
>
> We want to thank the reviewer for the constructive feedback. We have now added more datasets and baselines, and we refer to our answers below where we address the reviewer comments in detail.
>
> > W1. The experiments were run on only 4 fairly small datasets.
>
> Next to Cora-ML, Citeseer, Cora, and Pubmed, our paper also includes results on WikiCS and the *significantly larger* arXiv dataset in Figure 12 on Page 15. Furthermore, we now include label poisoning experiments on Amazon Photos and Amazon Computers, and we share the new results here https://figshare.com/s/ce37834b03144a1bf940 (Figure 1). Last but not least, we now also include stochastic block models, where we performed a systematic study on the effect of homophily (see Figures 4 & 5 in the link).
>
> Thus, our study now includes a total of **nine datasets**.
>
> > Q2 & W2. Section Related Works mentions multiple graph-domain certifiable defenses. The authors claim that none of them is applicable to this setting (each of the works is either not applicable to the perturbation models, or infinite-width GNNs, or scaling only to very small datasets). As a result, the authors only compare their method to an MLP baseline, which seems very simple. I would double-check if any of the related works can be extended or adapted to compare the results.
> > Q2. Can you provide a comparison with other works' baselines and elaborate further on why they are not adaptable? Even running and seeing a very poor result compared to this work would be valuable if verified experimentally.
>
> We now include **two baselines from previous work**. For structure certifications, we already compare to PGNNCert [1] in Appendix E Page 19 and *significantly outperform* them. We now also compare our label certificate to the infinite-width GNN certificates from [2] in Table 1. Initially, we did not compare to [2], as their method only works well for binary datasets having 20-40 labels *in total*. Given this sub-optimal setting for label partitioning and that the certificates from [2] are exact for infinite-width GNNs, we find it remarkable that our method provides comparable results for a perturbation budget of 5% and outperforms [2] for 10% perturbations. Note that our certificate scales to graph datasets several orders of magnitude larger in size (number of nodes) and number of labels than [2].
>
> Table 1: Cora-ML (binary) [2], Certified Accuracy [%]
>
> | Label Perturbations | 5% | 10% |
> | --- | --- | --- |
> | Ours (co-train then self-train) | 66.5 ± 17.8 | 36.21 ± 1.4 |
> | LabelCert [2] | 68.7 ± 6.8 | 32.6 ± 2.0 |
>
>
> [1] Li et al. "Deterministic certification of graph neural networks against graph poisoning attacks with arbitrary perturbations", CVPR 2025.
> [2] Sabanayagam et al. "Exact Certification of (Graph) Neural Networks Against Label Poisoning", ICLR 2025
>
> > Q1. The paper mentions that "our insights ... may be of interest beyond graph learning". Can you provide some concrete examples for applications (or potential experiments to run) for different domains?
>
> We are happy to provide concrete examples! In particular, self-training could be leveraged to strengthen partition-based certificates developed for the image domain, such as Semi-Supervised DPA developed by Levine & Feizi (2021) [1], and follow-up works on the partition-based certificate if applied to certify label poisoning [2,3]. All of which have not investigated self-training to strengthen the certificates, even though [1] partitions the label set for label poisoning certificates and the mechanisms in [2,3] could also be applied to certify label poisoning.
>
> [1] Levine & Feizi "Deep partition aggregation: Provable defense against general poisoning attack", ICLR 2021.
> [2] Wang et al. "Improved certified defenses against data poisoning with (deterministic) finite aggregation", ICML 2022.
> [3] Rezaei et al. "Run-off election: Improved provable defense against data poisoning attack", ICML 2023.

---

> > ### Author Rebuttal · Reviewer_hbbX · 2026-04-02
> >
> > Thank you. I maintain my positive score.

---

> > > ### Author Response · Authors · 2026-04-08
> > >
> > > We want to thank the reviewer again for their constructive review and for acknowledging that their concerns have been adequately addressed!

---

### Official Review · Reviewer_tR59 · 2026-03-13

**Soundness:** 3
**Presentation:** 3
**Significance:** 3
**Originality:** 2
**Overall Recommendation:** 4
**Confidence:** 4

**Summary:**

The paper proposes Self-Training Graph Partition Aggregation (ST-GPA). ST-GPA is a certified robustness framework for graph poisoning based on partitioning the graph into disjoint subsets and training and aggregating the sub-models on them. It studies the challenges of extending pertaining based robustness method from images (iid) to graphs (non-iid). First, it resolves the issue in semi-supervised learning, and then it proposes self-training and co-training to address the issues of sparse labels and links.

**Compliance With Llm Reviewing Policy:**

Affirmed.

**Final Justification:**

The authors provide several new experimental result which convinced me to raise my score.

**Key Questions For Authors:**

1. How tight are the certificates relative to empirical robustness under strong adaptive attacks?
2. Does the method remain effective under denser or differently structured graphs such as Amazon-Photo, Amazon-Computers

**Limitations:**

Partially: They discuss that their method depend on graph homophily.

**Strengths And Weaknesses:**

## Strengths

The paper addresses an interesting problem: obtaining robustness guarantees for graph-based models under label poisoning or structural perturbations.

The paper correctly explores all of the shortcomings of the partitioning method for graph poisoning (or, more generally, the semi-supervised learning setting).

The method seems to provide stronger certificates than a simpler non-co-training baseline, which suggests that the proposed training strategy has some value.



## Weaknesses


1- The partitioning component does not seem fundamentally new, and the core empirical gains appear to come from self-training or co-training across sub-models. Since similar ideas have been explored before on robustness [1] aside the next concern in practicality of the method it make its contribution limited.

2- Although the method seems to improve over non-co-training in terms of certified robustness, the guarantee still appears weak in practice on at least some datasets. In particular, based on Figure 12(b), the certified performance on OGBN-arXiv seems to degrade under a tiny perturbation budget which approaches MLP-level performance (55% based on the OBGN-arxiv leaderboard). particularly based on figure ~15 edges out of 1,166,243 is enough to break it certificate to MLP level performance.

3- It would be nice to see the gap between certified robustness and actual robustness under a strong attack for understanding whether the certificate is meaningful or overly conservative. I suggest PRBCD which explored for poisoning in [2].

Minor issues:

- In several figure (for example figure 4) the performance “Clean accuracy w/o partitioning” show as vertical line at zero perturbation and horizontal line for higher budget which can misinterpret that model performance is zero even with 1 poisoning edge budget.

- Reproducibility: The paper does not provide code.

[1] Li, Kuan, et al., Revisiting graph adversarial attack and defense from a data distribution perspective. ICLR 2023

[2] Mujkanovic, Felix, et al. "Are defenses for graph neural networks robust?." Advances in Neural Information Processing Systems 35 (2022)

---

> ### Author Rebuttal · Authors · 2026-03-31
>
> We want to thank the reviewer for the constructive feedback! We now include several new experimental results and datasets, and below address all the raised points in detail.
>
> > Q1 & W3. Tightness.
>
> We are currently investigating the tightness of our method through transferring the robustness benchmark developed in [2]. We will update our collection of new results (https://figshare.com/s/ce37834b03144a1bf940) in the coming days as soon as we have results. Note that PR-BCD from [2], and gradient-based attacks in general, can't be immediately employed for ensemble-based methods. This is caused by the fact that to differentiate through these methods w.r.t. the data, one would need to make the partitioning itself differentiable and then unroll $k$ many trainings of $k$ different models on $k$ different datasets (with $k$ often bigger $100$). We are not aware of any work that has shown how to overcome the resulting technical challenges in the differentiation yet. Thus, we opt for a transfer attack from [2]. We want to make the reviewer aware of [3], which shows for image datasets that DPA-based methods are overly conservative for small $k$, but usually tight for larger $k$, which more closely aligns with the $k$-choices in our work.
>
> [2] Mujkanovic et al. "Are defenses for graph neural networks robust?." NeurIPS 2022
> [3] Mohgaonkar et al. https://openreview.net/forum?id=mLLAc1FhWV
>
> > Q2. Denser / differently structured graphs. Amazon-Photo, Amazon-Computers
>
> Yes, the label poisoning certificate does remain effective for denser and differently structured graphs.  **We provide the positive results for Amazon-Photo and Amazon-Computers** in Figure 1 in https://figshare.com/s/ce37834b03144a1bf940.
>
> We acknowledge that for these comparatively dense graphs (average degree ~30–40), our proposed structure certificate is less effective than on the sparse datasets that we consider (average degree ~3–4). However, note that our method is particularly tailored to solve the challenging problem for sparse graphs, where it performs strongly. We believe this constitutes an important contribution, in particular since prior approaches were not able to provide non-trivial guarantees for either setting. In response to your comment, we will provide a more detailed discussion of the impact of graph density in the camera-ready version of the paper.
>
> We also want to note that we now provide new results where we systematically investigate and quantify the dependency of our method on homophily through sampling different graphs from contextual stochastic block models (Figures 4 & 5 in the above link). There, we vary homophily from 0% (pure heterophily) to 100% (pure homophily). We find that both our certificates against label and structure perturbations start to provide benefits from 60% homophily, with strong results for >70% homophily (exemplary, Cora has 81% homophily).
>
> > W1. Similar ideas explored before.
>
> While self-training has indeed been explored in the general robustness literature, the context, objectives, and approaches differ substantially from ours. Critically, it is unclear how such methods could address the sparsity challenges inherent to certifiable robustness in graphs. In contrast, we propose concrete mechanisms to overcome these limitations, enabling strong guarantees under sparse graph regimes. We believe this is a significant contribution that enables promising directions for future work.
>
> > W2. Weak on obgn-arxiv, MLP baseline.
>
> We want to note that Figure 12b refers to label und not structure poisoning. Thus, an MLP does not provide a valid baseline as it depends on the poisoned labels. (Indeed, the cited MLP will have a certified accuracy of 0% starting with only 1 label perturbation.) For a correct comparison, we have now computed a certificate for an ensemble of MLP's w/o our semi-supervised learning and find that it performs significantly inferior to our certificate, struggling to learn given the sparse labels in each partition. Further, we have not tuned the certificates' hyperparameters for arXiv. We have rerun the same experiment, increasing $k$ from $400$ to $800$ and achieve notable increases in certified accuracy. The detailed results can be found in Figure 2 in https://figshare.com/s/ce37834b03144a1bf940.
>
> > M1. “Clean accuracy w/o partitioning”
>
> The plots show on y-axis the certified accuracy, which for the clean model is 0% for any perturbation >0. We spent some time thinking about how to best represent the clean baseline and decided against "a straight line" as it could be misinterpreted as providing certified accuracy > 0, but we still wanted to highlight clean accuracy w/o partitioning. However, we feel that the label we gave here is suboptimal and we would propose calling it "Clean model w/o partitioning". We are also happy for other suggestions how to best represent the clean accuracy w/o partitioning.
>
> > M2. Code
>
> We do share the anonymized code in footnote 3 on Page 6 Lines 327-328.

---

> > ### Author Rebuttal · Reviewer_tR59 · 2026-04-03
> >
> > Thanks for rebuttal. I will raise my score.

---

> > > ### Author Response · Authors · 2026-04-08
> > >
> > > We want to thank the reviewer for raising their score and for acknowledging that their concerns have been adequately addressed!

---

### Decision · Program_Chairs · 2026-04-30

**Decision:**

Accept (regular)

**Comment:**

This paper proposes a new method named Self-Training Graph Partition Aggregation (ST-GPA). The proposed method aims to incorporate graph partition, pseudo-labeling, and synthetic edges for robustness certification. The proposed method is evaluated across different datasets and architectures.

After the rebuttal, all the reviewers gave positive ratings for the paper. The reviewers' comments, such as novelty concerns, performance explanation, and more baselines, have been resolved by the authors. Therefore, I recommend the acceptance of this paper.

Please include all the rebuttal content, such as additional experiments, in the final revision.